# Wide Neural Networks Trained with Weight Decay Provably Exhibit Neural Collapse

**Arthur Jacot**,[*]  **Peter Súkeník**,[†]  **Zihan Wang**,[‡]  **and Marco Mondelli**[§]

## Abstract

Deep neural networks (DNNs) at convergence consistently represent the training data in the last layer via a geometric structure referred to as neural collapse. This empirical evidence has spurred a line of theoretical research aimed at proving the emergence of neural collapse, mostly focusing on the unconstrained features model. Here, the features of the penultimate layer are free variables, which makes the model data-agnostic and puts into question its ability to capture DNN training. Our work addresses the issue, moving away from unconstrained features and studying DNNs that end with at least two linear layers. We first prove generic guarantees on neural collapse that assume *(i)* low training error and balancedness of linear layers (for within-class variability collapse), and *(ii)* bounded conditioning of the features before the linear part (for orthogonality of class-means, and their alignment with weight matrices). The balancedness refers to the fact that $W_{\ell+1}^\top W_{\ell+1} \approx W_\ell W_\ell^\top$ for any pair of consecutive weight matrices of the linear part, and the bounded conditioning requires a well-behaved ratio between largest and smallest non-zero singular values of the features. We then show that such assumptions hold for gradient descent training with weight decay: *(i)* for networks with a wide first layer, we prove low training error and balancedness, and *(ii)* for solutions that are either nearly optimal or stable under large learning rates, we additionally prove the bounded conditioning. Taken together, our results are the first to show neural collapse in the end-to-end training of DNNs.

## 1 Introduction

Among the many possible interpolators that a deep neural network (DNN) can find, Papyan et al. (2020) showed a strong bias of gradient-based training towards representations with a highly symmetric structure in the penultimate layer, which was dubbed *neural collapse* (NC). In particular, the feature vectors of the training data in the penultimate layer collapse to a single vector per class (NC1); these vectors form orthogonal or simplex equiangular tight frames (NC2), and they are aligned with the last layer's row weight vectors (NC3). The question of why and how neural collapse emerges has been considered by a popular line of research, see e.g. Lu & Steinerberger (2022); E & Wojtowytsch (2022) and the discussion in Section 2. Many of these works focus on a simplified mathematical framework: the unconstrained features model (UFM) (Mixon et al., 2020; Han et al., 2022; Zhou et al., 2022a), corresponding to the joint optimization over the last layer's weights and the penultimate layer's feature representations, which are treated as free variables. To account for the existence of the training data and of all the layers before the penultimate (i.e., the backbone of the network), some form of regularization on the free features is usually added. A number of papers has proved the optimality of NC in this model (Lu & Steinerberger, 2022; E & Wojtowytsch, 2022), its emergence with gradient-based methods (Mixon et al., 2020; Han et al., 2022) and a benign loss landscape (Zhou et al., 2022a; Zhu et al., 2021). However, the major drawback of the UFM lies in its data-agnostic nature: it only acknowledges the presence of training data and backbone through a simple form of regularization (e.g., Frobenius norm or sphere constraint), which is far from being equivalent to end-to-end training. Moving beyond UFM, existing results are either only applicable to rather shallow networks (of at most three layers) (Kothapalli & Tirer, 2024; Hong & Ling, 2024) or hold under strong assumptions, such as symmetric quasi-interpolation (Xu et al., 2023; Rangamani & Banburski-Fahey, 2022), block-structured empirical NTK throughout training (Seleznova et al., 2023), or geodesic structure of the features across all layers (Wang et al., 2024).

---

[*]Courant Institute of Mathematical Sciences, NYU. Email: `arthur.jacot@nyu.edu`

[†]Institute of Science and Technology Austria. Email: `peter.sukenik@ista.ac.at`

[‡]Courant Institute of Mathematical Sciences, NYU. Email: `zw3508@nyu.edu`

[§]Institute of Science and Technology Austria. Email: `marco.mondelli@ist.ac.at`

In this paper, we provide the first end-to-end proof of within-class variability collapse (NC1) for a class of networks that end with at least two linear layers. Furthermore, we give rather weak sufficient conditions – either near-optimality or stability under large learning rates – for solutions to exhibit the orthogonality of class-means (NC2) and the alignment of class means with the last weight matrix (NC3). More precisely, our contributions can be summarized as follows:

- First, we show that within-class variability collapse (i.e., NC1) occurs, as long as the training error is low and the linear layers are approximately balanced, i.e., $\left\|W_{\ell+1}^\top W_{\ell+1} - W_\ell W_\ell^\top\right\|_F$ is small, where $W_{\ell+1}, W_\ell$ are two consecutive weight matrices of the linear head. If, additionally, the conditioning of the linear head mapping is bounded, we bound the conditioning of the matrix of class means in the last layer, as well as that of the last weight matrix. This implies that, as the number of linear layers grows, class means become orthogonal and, furthermore, they align with the last layer's row vectors, which proves both NC2 and NC3.

- Next, we show that the sufficient conditions above for NC1 are satisfied by a class of deep networks with a wide first layer, smooth activations and pyramidal topology, after gradient training with weight decay. This provides the first guarantee of the emergence of NC1 for a deep network trained end-to-end via gradient descent.

- We present two sufficient conditions under which the linear head is well-conditioned, hence NC1, NC2 and NC3 hold: either the network approaches a global optimum of the $\ell_2$-regularized square loss, or it nearly interpolates the data while being stable under large learning rates.

- Our numerical experiments on various architectures (fully connected, ResNet) and datasets (MNIST, CIFAR) confirm the insights coming from the theory: *(i)* NC2 is more prominent as the depth of the linear head increases, and *(ii)* the final linear layers are balanced at convergence. Furthermore, we show that, as the non-linear part of the network gets deeper, the non-negative layers become less non-linear and more balanced.

## 2 RELATED WORK

**Neural collapse.** Since its introduction by Papyan et al. (2020), neural collapse has been intensively studied, both from a theoretical and practical viewpoint. Practitioners use the NC for a number of applications, including transfer learning, OOD detection and generalization bounds (Galanti et al., 2022a; Haas et al., 2022; Ben-Shaul & Dekel, 2022; Li et al., 2023a;b; Zhang et al., 2024). On the theoretical front, the most widely adopted framework to study the emergence of NC is the unconstrained features model (UFM) (Mixon et al., 2020; Fang et al., 2021). Under the UFM, the NC has been proved to be optimal with cross-entropy loss (E & Wojtowytsch, 2022; Lu & Steinerberger, 2022; Kunin et al., 2022), MSE loss (Zhou et al., 2022a) and other losses (Zhou et al., 2022b). Optimality guarantees for the generalization of NC to the class-imbalanced setting have been provided by Fang et al. (2021); Thrampoulidis et al. (2022); Hong & Ling (2023); Dang et al. (2024). Besides global optimality, a benign loss landscape around NC solutions has been proved in Zhu et al. (2021); Ji et al. (2022); Zhou et al. (2022a), and the emergence of NC with gradient-based optimization under UFM has been studied by Mixon et al. (2020); Han et al. (2022); Ji et al. (2022); Wang et al. (2022). Jiang et al. (2023) extend the analysis to large number of classes, Kothapalli et al. (2023) to graph neural networks, Tirer et al. (2023) generalize UFM with a perturbation to account for its imperfections, while Andriopoulos et al. (2024) generalize UFM and NC to regression problems. A deep neural collapse is theoretically analyzed with deep UFM for the linear case in Dang et al. (2023); Garrod & Keating (2024), for linear case with data-specific assumptions in Wang et al. (2023), for the non-linear case with two layers in Tirer & Bruna (2022), and for the deep non-linear case in Súkeník et al. (2023; 2024), where, notably, the latter work provides the first negative result on deep NC with deep UFM.

A line of recent work aims at circumventing the data-agnostic nature of UFM, showing the emergence of neural collapse in settings closer to practice. Specifically, Seleznova et al. (2023) assume a block structure in the empirical NTK matrix. Kernels are used by Kothapalli & Tirer (2024) to analyze NC in wide two-layer networks, showing mostly negative results in the NTK regime. Beaglehole et al. (2024) prove the emergence of deep neural collapse using kernel-based layer-wise training, and they show that NC is an optimal solution of adaptive kernel ridge regression in an over-parametrized regime. Sufficient (but rather strong) conditions for the emergence of NC beyond UFM are provided by Pan & Cao (2023). Hong & Ling (2024) analyze two and three layer networks end-to-end, but only provide conditions under which the UFM optimal solutions are feasible.

Wang et al. (2024) consider the NC formation in residual networks, and the model is similar to the perturbed UFM in Tirer et al. (2023). However, the results crucially assume that the features lie on a geodesic in euclidean space, and proving that this is the case after training the ResNet is an open problem. Rangamani & Banburski-Fahey (2022); Xu et al. (2023) focus on homogeneous networks trained via gradient-based methods, making the strong assumption of symmetric quasi-interpolation. Then, Rangamani & Banburski-Fahey (2022) do not prove this assumption, and the argument of Xu et al. (2023) requires a regularization different from the one used in practice as well as interpolators with a given norm (whose existence is an open question). We also note that the quasi-interpolation property does not hold in practice exactly, which points to the need of a perturbation analysis.

**Implicit bias.** Our approach leverages the (approximate) balancedness of the weights, which plays a central role in the analysis of the training dynamics of linear networks (Arora et al., 2018b) and leads to a bias towards low-rank matrices (Arora et al., 2019; Tu et al., 2024). In the presence of weight decay, the low-rank bias can be made even more explicit (Dai et al., 2021), and it is reinforced by stochastic gradient descent (Wang & Jacot, 2024). Moving towards nonlinear models, shallow networks with weight decay exhibit low-rank bias, as described by the variational norm (Bach, 2017), or Barron norm (E et al., 2019), and this bias provably emerges under (modified) GD dynamics (Abbe et al., 2022; Bietti et al., 2022; Lee et al., 2024). However, a single hidden layer appears to be insufficient to exhibit NC for most datasets. Moving towards deep nonlinear models, networks with weight decay are also known to exhibit low-rank bias (Galanti et al., 2022b; Jacot et al., 2022), which can be related to neural collapse (Zangrando et al., 2024). The dimensionality and rank of the weights/representations varies between layers, exhibiting a bottleneck structure (Jacot, 2023a;b; Wen & Jacot, 2024). We remark that existing results apply to the global minima of the $\ell_2$-regularized loss. In contrast, our paper provides rather general sufficient conditions that are then provably satisfied by GD, thus showing that the training algorithm is responsible for neural collapse.

## 3 BALANCEDNESS AND INTERPOLATION IMPLY NEURAL COLLAPSE

**Notation and problem setup.** Given a matrix $A$ of rank $k$, we denote by $A_{i:}$ its $i$-th row, by $A_{:i}$ its $i$-th column, by $s_1(A) \geq \cdots \geq s_k(A)$ its singular values in non-increasing order, and by $\kappa(A)$ the ratio $\frac{s_1(A)}{s_k(A)}$ between its largest and smallest non-zero singular values. We denote by $\|A\|_F, \|A\|_{op}$ and $\sigma_{\min}(A)$ its Frobenius norm, its operator norm and its smallest singular value, respectively.

We consider a neural network with $L_1$ non-linear layers with activation function $\sigma : \mathbb{R} \to \mathbb{R}$ followed by $L_2$ linear layers. Let $L := L_1 + L_2$ be the total number of layers, $W_\ell \in \mathbb{R}^{n_\ell \times n_{\ell-1}}$ the weight matrix at layer $\ell$, $X \in \mathbb{R}^{d \times N}$ the training data and $Y \in \mathbb{R}^{K \times N}$ the labels (for consistency, we set $n_0 = d$ and $n_L = K$), where the output dimension $K$ corresponds to the number of classes and $N$ is the number of samples. We consider a one-hot encoding, i.e., the rows of $Y$ are elements of the canonical basis. Let $Z_\ell \in \mathbb{R}^{n_\ell \times N}$ be the output of layer $\ell$, given by

$$Z_\ell = \begin{cases} X & \ell = 0, \\ \sigma(W_\ell Z_{\ell-1}) & \ell \in [L_1], \\ W_\ell Z_{\ell-1} & \ell \in \{L_1+1, \ldots, L_1+L_2\}, \end{cases} \tag{1}$$

where the activation function $\sigma$ is applied componentwise and, given an integer $n$, we use the shorthand $[n] := \{1, \ldots, n\}$. We write $\theta$ for the vector obtained by concatenating all parameters $\{W_i\}_{i \in [L]}$ and $W_{m:\ell}$ for the partial products of the weight matrices $W_m \cdots W_\ell$, so that $Z_m = W_{m:\ell+1} Z_\ell$ for all $m \geq \ell \in \{L_1+1, \ldots, L_1+L_2\}$. We index individual samples in a feature matrix $Z$ as $z_{ci}$, meaning the $i$-th sample of the $c$-th class, sometimes adding an upper-index to denote the layer or matrix to which the sample belongs. Let $\mu_c$ denote the mean of all samples from class $c$ and $\mu_G$ the global mean. Let $\bar{Z}$ be the matrix of class-means stacked into columns. The NC1 metric on a feature matrix $Z$ is given by $\frac{\text{tr}(\Sigma_W)}{\text{tr}(\Sigma_B)}$, where $\Sigma_W = \frac{1}{N} \sum_{c,i} (z_{ci} - \mu_c)(z_{ci} - \mu_c)^\top$ and $\Sigma_B = \frac{1}{K} \sum_{c=1}^K (\mu_c - \mu_G)(\mu_c - \mu_G)^\top$. The NC2 metric on a feature matrix $Z$ is defined as $\kappa(\bar{Z})$, i.e., the conditioning number of the class-mean matrix of $Z$. The NC3 metric on a feature matrix $Z$ and a weight matrix $W$ is defined as $\frac{1}{N} \sum_{c,i} \cos(z_{ci}, W_{c:})$, i.e., the average cosine similarity of features and weight vectors corresponding to the features' class.

At this point, we state our result giving a set of sufficient conditions for NC1, NC2 and NC3.

**Theorem 3.1.** *If the network satisfies*

- *approximate interpolation, i.e., $\|Z_L - Y\|_F \leq \epsilon_1$,*

- *approximate balancedness, i.e.,* $\left\|W_{\ell+1}^\top W_{\ell+1} - W_\ell W_\ell^\top\right\|_{op} \leq \epsilon_2$, *for* $\ell \in \{L_1 + 1, \ldots, L - 1\}$,

- *bounded representations and weights, i.e.,* $\|Z_{L-2}\|_{op}, \|Z_{L-1}\|_{op}, \|W_\ell\|_{op} \leq r$, *for* $\ell \in \{L_1 + 1, \ldots, L\}$,

*then if* $\epsilon_1 \leq \min\left(s_K(Y), \sqrt{\frac{(K-1)N}{4K}}\right)$,

$$NC1(Z_{L-1}) = O\left((\epsilon_1 + \sqrt{\epsilon_2})^2\right). \tag{2}$$

*If we additionally assume that the linear part of the network is not too ill-conditioned, i.e.,* $\kappa(W_{L:L_1+1}) \leq c_3$, *then*

$$\kappa(W_L) \leq c_3^{\frac{1}{L_2}} + O(\epsilon_2), \tag{3}$$

$$NC2(Z_{L-1}) \leq c_3^{\frac{1}{L_2}} + O(\epsilon_1 + \sqrt{\epsilon_2}), \tag{4}$$

$$NC3(Z_{L-1}, W_L) \geq \frac{c_3^{-\frac{1}{L_2}} + c_3^{-\frac{3}{L_2}}}{2} - O\left(\epsilon_1 + \left(\sqrt{\epsilon_2} + c_3^{\frac{1}{L_2}} - 1\right)^2\right). \tag{5}$$

The restrictions $\epsilon_1 \leq s_K(Y)$ and $\epsilon_1 \leq \sqrt{\frac{(K-1)N}{4K}}$ are mild, and we are interested in the regime in which $\epsilon_1$ is small. In words, (2) shows that, when $\epsilon_1, \epsilon_2 \approx 0$ (i.e., the network approximately interpolates the data in a balanced way), the within-class variability (which captures NC1) vanishes. If in addition the depth of the linear part of the network grows, the RHS of (3) approaches 1, i.e., the last weight matrix $W_L$ is close to orthogonal. This implies that *(i)* $Z_{L-1}$ is also close to orthogonal (which captures NC2), and *(ii)* the weights in the last layer align with $Z_{L-1}$ (which captures NC3). In fact, when $\epsilon_1, \epsilon_2 \approx 0$ and $\kappa(W_L) \approx 1$, the RHS of (4) and (5) is close to 1. Below we give a proof sketch. The complete argument is deferred to Appendix B, where we state a more precise version of the result (see Theorem B.1) tracking the dependence of the bounds on $r, N, K$ and $s_K(Y)$.

**Proof sketch.** We start with NC1. If $Z_L$ is already well-collapsed (which is guaranteed by the approximate interpolation) and $W_L$ is well-conditioned, then the only source of within-class variability in $Z_{L-1}$ is within the null space of $W_L$. However, if $W_L$ and $W_{L-1}$ are balanced, the image of $Z_{L-1}$ must be approximately in a subspace of the row space of $W_L$ and, hence, $Z_{L-1}$ has little freedom within the kernel of $W_L$. More formally, consider first the case of perfect balancedness (i.e., $\epsilon_2 = 0$), and denote by $W_L^+$ the pseudo-inverse of $W_L$. Then, $\text{Im}(Z_{L-1}) \subset \text{Im}(W_L^\top)$ and

$$\left\|W_L^+ W_L Z_{L-1} - W_L^+ Y\right\|_F = \left\|Z_{L-1} - W_L^+ Y\right\|_F \leq \frac{\epsilon_1}{s_K(W_L)}. \tag{6}$$

As $s_K(W_L)$ can be lower bounded by using the assumptions on approximate interpolation and boundedness of representations, the RHS of (6) is small and, therefore, $Z_{L-1}$ is close to a matrix with zero within-class variability. Moving to the case $\epsilon_2 \neq 0$, we need to show that $W_L^+ W_L Z_{L-1}$ is close to $Z_{L-1}$. As $W_L^+ W_L$ projects onto the row-space of $W_L$, only the part of $Z_{L-1}$ in the kernel of $W_L$ has to be considered. This part is controlled after writing $Z_{L-1} = W_{L-1} Z_{L-2}$, using the boundedness of $Z_{L-2}$ and the approximate balancedness between $W_L$ and $W_{L-1}$. Finally, as $Z_{L-1}$ is close to $W_L^+ Y$, a direct computation yields the bound on NC1.

Next, to bound $\kappa(W_L)$, we notice that $(W_L W_L^\top)^{L_2} - (W_{L:L_1+1} W_{L:L_1+1}^\top)$ has small operator norm, since the weights of linear layers are approximately balanced. This allows to upper bound $\kappa(W_L)^{2L_2}$ in terms of $\kappa(W_{L:L_1+1})$ (plus a small perturbation), which gives (3). To lower bound the NC3 metric, we rescale $Z_{L-1}$ and $W_L$ to $Z'_{L-1}$ and $W'_L$, so that their columns and rows, respectively, have roughly equal size. Then, we reformulate the problem to proving that $\langle Z'_{L-1}, W'_L Y \rangle$ is close to its theoretical maximum. We proceed to show this by arguing that, in this scaling and given that $W_L$ is sufficiently well-conditioned, $W'_L$ can be replaced by $(W'_L)^+$. As $Z_{L-1}$ is close to $W_L^+ Y$, we obtain (5). Finally, the bound on NC2 in (4) follows by combining (3) with the closeness between $Z_{L-1}$ and $W_L^+ Y$ already obtained in the proof of NC1.

## 4    Gradient Descent Leads to No Within-Class Variability (NC1)

In this section, we show that NC1 holds for networks with one wide layer followed by a pyramidal topology, as considered in Nguyen & Mondelli (2020). To do so, we show that the balancedness and

interpolation conditions of Theorem 3.1 hold. We consider the neural network in (1) and minimize the $\lambda$-regularized square loss $C_\lambda(\theta) = \frac{1}{2}\|z_L(\theta) - y\|_2^2 + \frac{\lambda}{2}\|\theta\|_2^2$, where $z_L$ and $y$ are obtained by vectorizing $Z_L$ and $Y$, and $\theta$ collects all the network parameters. To do so, we consider the gradient descent (GD) update $\theta_{k+1} = \theta_k - \eta\nabla C_\lambda(\theta_k)$, where $\eta$ is the step size and $\theta_k = (W_\ell^k)_{\ell=1}^L$ contains all parameters at step $k$. We also denote by $Z_\ell^k$ the output of layer $\ell$ after $k$ steps of GD. We make the following assumption on the pyramidal topology of the network, noting that this requirement is also common in prior work on the loss landscape (Nguyen & Hein, 2017; 2018).

**Assumption 4.1.** *(Pyramidal network topology) Let $n_1 \geq N$ and $n_2 \geq n_3 \geq \ldots \geq n_L$.*

We make the following assumptions on the activation function $\sigma$ of the non-linear layers.

**Assumption 4.2.** *(Activation function) Fix $\gamma \in (0,1)$ and $\beta \geq 1$. Let $\sigma$ satisfy that: (i) $\sigma'(x) \in [\gamma, 1]$, (ii) $|\sigma(x)| \leq |x|$ for every $x \in \mathbb{R}$, and (iii) $\sigma'$ is $\beta$-Lipschitz.*

This includes smooth leaky ReLUs, the same assumption was employed by Nguyen & Mondelli (2020); Frei et al. (2022) and a similar one by Chatterjee (2022). In principle, $\sigma$ can change at all layers, as long as it satisfies the above requirement. Next, let us introduce some notation[1] for the singular values of the weight matrices at initialization $\theta_0 = (W_\ell^0)_{\ell=1}^L$:

$$\lambda_\ell = \sigma_{\min}(W_\ell^0), \quad \bar{\lambda}_\ell = \|W_\ell^0\|_{op} + \min_{\ell \in \{3,\ldots,L\}} \lambda_\ell, \quad \lambda_{i \to j} = \prod_{\ell=i}^j \lambda_\ell, \quad \bar{\lambda}_{i \to j} = \prod_{\ell=i}^j \bar{\lambda}_\ell. \quad (7)$$

We also define $\lambda_F = \sigma_{\min}(\sigma(W_1^0 X))$ as the smallest singular value of the output of the first hidden layer at initialization. Finally, we make the following assumption on the initialization.

**Assumption 4.3.** *(Initial conditions)*

$$\lambda_F \lambda_{3 \to L} \min(\lambda_F, \min_{\ell \in \{3,\ldots,L\}} \lambda_\ell) \geq 8\gamma\sqrt{\left(\frac{2}{\gamma}\right)^L C_0(\theta_0)}. \quad (8)$$

We note that (8) can be satisfied by choosing a sufficiently small initialization for the second layer and a sufficiently large one for the remaining layers. In fact, the LHS of (8) depends on all the layer weights except the second, so this quantity can be made arbitrarily large. Next, by taking a sufficiently small second layer, the term $\sqrt{2C_0(\theta)} = \|Z_L - Y\|_F$ can be upper bounded by $2\|Y\|_F$. As a consequence, the RHS of (8) is at most $8\sqrt{2}\|Y\|_F \gamma \left(\frac{2}{\gamma}\right)^{L/2}$. As the LHS of (8) can be arbitrarily large, the inequality holds for a suitable initialization.

**Theorem 4.4.** *Let the network satisfy Assumption 4.1, $\sigma$ satisfy Assumption 4.2 and the initial conditions satisfy Assumption 4.3. Fix $0 < \epsilon_1 \leq \frac{1}{2}\sqrt{\frac{(K-1)N}{K}}$, $\epsilon_2 > 0$, and run $k$ steps of $\lambda$-regularized GD with step size $\eta$, where $\lambda = \Theta(\epsilon_1^2)$, $\eta = O(\epsilon_2)$ and $k = \Omega\left(\frac{1}{\epsilon_1^2 \epsilon_2}\log\frac{1}{\epsilon_2}\right)$. Then,*

$$NC1(Z_{L-1}^k) = O\left((\epsilon_1 + \sqrt{\epsilon_2})^2\right). \quad (9)$$

In words, if regularization and learning rate are small enough and we run GD for sufficiently long, then the within-class variability vanishes. Below we provide a proof sketch. The complete argument is deferred to Appendix B, where we state a more precise version of the result (see Theorem B.2) tracking the dependence of $\lambda, \eta, k$ and $NC1(Z_{L-1}^k)$ on $N$, $K$, the data $X$, the network architecture and the initial conditions.

**Proof sketch.** We show that the network trained via $\lambda$-regularized GD fulfills the three sufficient conditions for NC1 given by Theorem 3.1, i.e., approximate interpolation, approximate balancedness and bounded representations/weights. To do so, we distinguish two phases in the training dynamics.

The *first phase* lasts for logarithmic time in $1/\lambda$ (or, equivalently, $1/\epsilon_1$) and, here, the loss decreases exponentially fast to a value of at most $2\lambda m_\lambda \leq \epsilon_1^2$. As the learning rate is small enough, the loss cannot increase during the GD dynamics, which already gives approximate interpolation. To show the exponential convergence, we proceed in two steps. First, Lemma 4.1 in (Nguyen & Mondelli, 2020) gives that the unregularized loss $C_0(\theta)$ satisfies the Polyak-Lojasiewicz (PL) inequality

$$\|\nabla C_0(\theta)\|_2^2 \geq \frac{\alpha}{2}C_0(\theta), \quad (10)$$

---

[1]To avoid confusion, we note that this notation is different from the one used in (Nguyen & Mondelli, 2020).

for all $\theta$ in a ball centered at initialization $\theta_0$ and with sufficiently large radius (captured by $r_0$). Next, we show that, if $C_0(\theta)$ satisfies the $\alpha$-PL inequality in (10), then the regularized loss $C_\lambda(\theta) = C_0(\theta) + \frac{\lambda}{2} \|\theta\|_2^2$ satisfies a shifted $\alpha$-PL inequality, which implies exponential convergence. This second step is formalized by the proposition below proved in Appendix B.

**Proposition 4.5.** *Let $C_0(\theta)$ satisfy the $\alpha$-PL inequality (10) in the ball $B(\theta_0, r_0)$. Then, in the same ball, $C_\lambda(\theta)$ satisfies the inequality*

$$\|\nabla C_\lambda(\theta))\|_2^2 \geq \frac{\alpha}{4} \left(C_\lambda(\theta) - \lambda m_\lambda\right), \tag{11}$$

*where $m_\lambda = (1 + \sqrt{4\lambda/\alpha})^2 (\|\theta_0\|_2 + r_0)^2$. Furthermore, assume that $r_0 \geq 8\sqrt{C_\lambda(\theta_0)/\alpha}$ and $\nabla C_0(\theta)$ is $\beta_1$-Lipschitz in $B(\theta_0, r_0)$. Then, for any $\eta < 1/(2\beta_1)$, there exists*

$$k_1 \leq \left\lceil \frac{\log \frac{\lambda m_\lambda}{C_\lambda(\theta_0) - \lambda m_\lambda}}{\log(1 - \eta \frac{\alpha}{8})} \right\rceil \tag{12}$$

*such that the $k_1$-th iterate of GD satisfies*

$$C_\lambda(\theta_{k_1}) \leq 2\lambda m_\lambda, \qquad \|\theta_{k_1} - \theta_0\|_2 \leq 8\sqrt{\frac{C_\lambda(\theta_0)}{\alpha}} \leq r_0. \tag{13}$$

The *second phase* lasts for linear time in $1/\lambda$ (or, equivalently in $1/\epsilon_1^2$) and logarithmic time in $1/\epsilon_2$ and, here, the weight matrices in the linear part of the network become balanced. More precisely, we adapt the analysis of (Du et al., 2018) to show that, if $W_\ell$ is a weight matrix of the linear part, $\|W_{\ell+1}^\top W_{\ell+1} - W_\ell W_\ell^\top\|_{op}$ decreases exponentially and the exponent scales with $1/\lambda$, which gives approximate balancedness. Finally, as the regularization term in the loss is at most $\epsilon_1^2$, the operator norm of representations and weight matrices is bounded by $r$ as in (29), and the proof is completed by an application of Theorem 3.1.

# 5 ORTHOGONALITY OF CLASS MEANS (NC2) AND ALIGNMENT WITH LAST WEIGHT MATRIX (NC3)

To guarantee the orthogonality of class means and their alignment with the last weight matrix, the crux is to show that the condition number $\kappa(W_L)$ of the last weight matrix $W_L$ is close to one. In fact, as $Z_{L-1} \approx W_L^+ Y$, this implies that the last hidden representation $Z_{L-1}$ is approximately a rotation and rescaling of $Y$, which gives NC2, and a bound on NC3 of the form in (5) also follows.

The fact that $\kappa(W_L) \approx 1$ is a consequence of the presence of many balanced linear layers. Indeed, balancedness implies $W_L W_L^\top = (W_{L:L_1+1} W_{L:L_1+1}^\top)^{\frac{1}{L_2}}$, which gives that $\kappa(W_L) = \kappa(W_{L:L_1+1})^{\frac{1}{L_2}}$. Thus, if the conditioning of the product of the linear layers $W_{L:L_1+1}$ can be bounded independently of $L_2$, one can guarantee that the conditioning of $W_L$ approaches 1 as $L_2 \to \infty$. As this is difficult to obtain in full generality (in particular the assumptions of Theorem 3.1 may not be sufficient), we show that the conditioning can be controlled *(i)* at any global minimizer, and *(ii)* when the parameters are 'stable' under large learning rates.

## 5.1 GLOBAL MINIMIZERS

We first show that any set of parameters that approximately interpolate with small norm has bounded condition number. We will then show that with the right choice of widths and ridge, all global minimizers satisfy these two assumptions.

**Proposition 5.1.** *Let $\sigma$ satisfy Assumption 4.2. Then, for any network that satisfies*

- *approximate interpolation, i.e., $\|Z_L - Y\|_F \leq \epsilon_1$,*

- *bounded parameters, i.e., $\|\theta\|_2^2 \leq LK + c$,*

*the linear part $W_{L:L_1+1}$ satisfies*

$$\kappa(W_{L:L_1+1}) \leq \exp\left(\frac{1}{2}\left(c + L_1 K \log K - 2K \log \frac{s_K(Y) - \epsilon_1}{\|X\|_{op}}\right)\right). \tag{14}$$

**Theorem 5.2.** *Let $\sigma$ satisfy Assumption 4.2. Assume there exist parameters of the nonlinear part $\theta_{nonlin} = (W_\ell)_{\ell=1}^{L_1}$ such that $Z_{L_1} = Y$ and $\|\theta_{nonlin}\|_2^2 = c$. Then, at any global minimizer of the regularized loss $\mathcal{L}_\lambda(\theta) = \frac{1}{2}\|Y - Z_L\|_F^2 + \frac{\lambda}{2}\|\theta\|_2^2$ with $\lambda \leq \frac{\epsilon_1^2}{KL+c}$, we have*

$$\kappa(W_{L:L_1+1}) \leq \left(\frac{\|X\|_{op}}{s_K(Y) - \epsilon_1}\right)^K \exp\left(\frac{1}{2}\left(c - L_1 K + L_1 K \log K\right)\right),$$

$$\kappa(W_L) \leq \left(\frac{\|X\|_{op}}{s_K(Y) - \epsilon_1}\right)^{\frac{K}{L_2}} \exp\left(\frac{1}{2L_2}\left(c - L_1 K + L_1 K \log K\right)\right). \tag{15}$$

*This implies that the bounds on NC1, NC2 and NC3 in (2), (4) and (5), respectively, hold with $\kappa(W_L)$ upper bounded as above and $\epsilon_2 = 0$.*

The assumption that the parameters of the nonlinear part can be chosen to fit the labels $Y$ is guaranteed for large enough width *(i)* by relying on any traditional approximation result (Hornik et al., 1989; Leshno et al., 1993; Arora et al., 2018a; He et al., 2018), or *(ii)* by taking the infinite time limit of any convergence results (Nguyen & Mondelli, 2020), or *(iii)* by taking the limit $\lambda \searrow 0$ in Proposition 4.5. A sketch of the arguments is below, with full proofs deferred to Appendix B.

**Proof sketch.** To prove Proposition 5.1, we write the norm of the linear and nonlinear parts of the network in terms of the conditioning of $W_{L:L_1+1}$: for the linear part, this is a direct computation; for the nonlinear part, we use Theorem 1 of Dai et al. (2021), which lower bounds the norm of the parameters in terms of the product of the singular values, and manipulate the latter quantity to obtain again the desired conditioning. Next, to prove Theorem 5.2, we pick the parameters of the nonlinear part $\theta_{nonlin}$ s.t. $Z_{L_1} = Y$ and $\|\theta_{nonlin}\|_2^2 = c$, and set the linear layers to the identity. This leads to a total parameter norm of $KL_2 + c$ and a regularized cost of $\frac{\lambda}{2}(KL_2 + c)$, and it forces the global minimizer to satisfy the assumptions of Proposition 5.1, which gives the claim on $\kappa(W_{L:L_1+1})$. Then, as all local minimizers have balanced linear layers, $\kappa(W_L) = \kappa(W_{L:L_1+1})^{\frac{1}{L_1}}$, which gives (15). Finally, the claim on NC1, NC2 and NC3 follows from an application of Theorem 3.1.

## 5.2 Large Learning Rates

Previous works have observed that the learning rates used in practice are typically 'too large', i.e. the loss may not always be strictly decreasing and GD diverges from gradient flow (Cohen et al., 2021). Thankfully, instead of simply diverging, for large $\eta$ (but not too large) the parameters naturally end up at the 'edge of stability': the top eigenvalue of the Hessian $\mathcal{H}C_\lambda$ is close to $\frac{2}{\eta}$, i.e., the threshold below which GD is stable (Cohen et al., 2021; Lewkowycz et al., 2020). One can thus interpret GD with learning rate $\eta$ as minimizing the cost $C_\lambda$ amongst parameters $\theta$ such that $\|\mathcal{H}C_\lambda\|_{op} < \frac{2}{\eta}$. These observations are supported by strong empirical evidence and have been also proved theoretically for simple models (Damian et al., 2022), although a general result remains difficult to prove due to the chaotic behavior of GD for large $\eta$. Specifically, the Hessian has the form

$$\mathcal{H}C_\lambda(\theta) = (\nabla_\theta Z_L)^\top \partial_\theta Z_L + \mathrm{Tr}\left[(Y - Z_L)\nabla_\theta^2 Z_L\right] + \lambda I_P, \tag{16}$$

with $\nabla_\theta Z_L \in \mathbb{R}^{P \times NK}$ and $P$ the number of parameters. The first term is the Fisher information matrix: this is dual to the NTK $\Theta = \nabla_\theta Z_L (\nabla_\theta Z_L)^\top \in \mathbb{R}^{NK \times NK}$ (Jacot et al., 2018). Therefore, at approximately interpolating points, we have $\|\mathcal{H}C_\lambda(\theta)\|_{op} = \|\Theta\|_{op} + O(\epsilon_1) + O(\lambda)$, where $\epsilon_1$ is the interpolation error and $\lambda$ the regularization parameter. We can thus interpret large learning rates as forcing a bound on the operator norm of the NTK. For networks that approximately interpolate the data with bounded NTK and bounded weights, the following proposition (proved in Appendix B) guarantees good conditioning of the weights in the linear part (and therefore NC2-3):

**Proposition 5.3.** *For any network that satisfies*

- *bounded NTK, i.e., $\|\Theta\|_{op} = \max_A \frac{\|\nabla_\theta \mathrm{Tr}[Z_L A^T]\|_2^2}{\|A\|_F^2} \leq CL_2$,*

- *approximate interpolation, i.e., $\|Z_L - Y\|_F \leq \epsilon_1$,*

- *bounded weights, i.e., $\|W_\ell\|_{op} \leq r$,*

*for any $M \leq L_2$, there is $\ell \in \{L_1 + 1, \ldots, L_1 + M\}$ such that $\kappa(W_{L:\ell}) \leq \frac{\sqrt{CL_2}Kr}{\sqrt{M}(s_K(Y) - \epsilon_1)}$.*

*Furthermore, any network that satisfies approximate interpolation and bounded weights is such that*

$$\|\Theta\|_{op} \geq \frac{(s_K(Y) - \epsilon_1)^2}{K^2 r^2} L_2. \tag{17}$$

As an example, by choosing $M = \frac{L_2}{2}$, we guarantee that there is at least one layer $\ell$ in the first half of the linear layers s.t. $\kappa(W_{L:\ell}) \leq \frac{\sqrt{2C}Kr}{s_K(Y) - \epsilon_1}$. Now, assuming the 'edge of stability' phenomenon, a learning rate of $\eta = \frac{\eta_0}{L_2}$ implies a bound $\|\Theta\|_{op} \leq \|\mathcal{H}C_\lambda\|_{op} + O(\epsilon_1) + O(\lambda) \leq \frac{2L_2}{\eta_0} + O(\epsilon_1) + O(\lambda)$, and thus Proposition 5.3 implies that there is a linear layer with bounded conditioning. The issue is that the proof of Theorem 4.4 requires an extremely small learning rate for large depths ($\eta \sim c^{-L}$, see Theorem B.2 in Appendix B), whereas Proposition 5.3 is only useful when $\eta \sim L^{-1}$. Note that the first phase of training (where the dynamics is analogous to the NTK regime) is stable when $\eta \sim L^{-1}$; it is the second phase that appears to require $\eta \sim c^{-L}$ to ensure GD remains close to gradient flow (we have much less control of the dynamics in this second phase and so we cannot rule out the possibility that GD approaches highly unstable regions). We are hopeful that a more careful analysis could show that a learning rate of $\eta \sim L^{-1}$ would push GD away from ill-conditioned and unstable regions (possibly deviating from gradient flow), so that at the end of the second phase the network would be interpolating, balanced, and have well-conditioned linear part thus guaranteeing NC1-3.

## 6 NUMERICAL RESULTS

In all experiments, we consider MSE loss and standard weight decay regularization. We train an MLP and a ResNet20 with an added MLP head on standard datasets (MNIST, CIFAR10), considering as backbone the first two layers for the MLP and the whole architecture before the linear head for the ResNet. We evaluate the following metrics related to neural collapse: for NC1, we compute $\mathrm{tr}(\Sigma_W)/\mathrm{tr}(\Sigma_B)$, where $\Sigma_W, \Sigma_B$ are the within- and between-class variability matrices of the feature matrices, respectively; for NC2, we display the conditioning number of the class-mean matrix; for NC3, we use the average cosine angle between the rows of a weight matrix and the columns of the preceding class-mean matrix. Finally, for balancedness we use $\frac{\left\|W_{\ell+1}^\top W_{\ell+1} - W_\ell W_\ell^\top\right\|_{op}}{\min\left\{\left\|W_{\ell+1}^\top W_{\ell+1}\right\|_{op}, \left\|W_\ell W_\ell^\top\right\|_{op}\right\}}$ and for negativity we use $\frac{\|Z_\ell - \sigma(Z_\ell)\|_{op}}{\|Z_\ell\|_{op}}$. We measure such metrics (and also index the layers) starting from the output of the backbone. Our findings can be summarized as follows (see Appendix A for additional complementary experiments).

**The deeper the linear head, the more clear NC occurs.** We first test whether the models with deep linear heads exhibit NC and if that's the case, whether it gets better as we deepen the linear head. In Figure 1, we show the NC metrics and the gram matrices of the class-mean matrices of the last layers for the training on CIFAR10 of ResNet20 with 6 extra layers of which the first three have a ReLU activation. We use weight decay of $0.001$ and learning rate of $0.001$, training for 5000 epochs (the learning rate drops ten-fold after 80% of the epochs in all our experiments). The plot clearly shows that the collapse is reached throughout the training. We also see that the NC2 metric improves progressively with each layer of the linear head, as predicted by our theory. This effect is also clearly visible from the gram matrices of the class-means (bottom row of Figure 1), which rapidly converge towards the identity. We note that these findings are remarkably consistent across a wide variety of hyperparameter settings and architectures. Wang et al. (2023) present similar results.

In Figure 2, we plot the dependence of the NC metrics on the number of layers in the linear head. We train an MLP on MNIST with 5 non-linear layers and a number of linear layers ranging from 1 to 5. We average over 5 runs per each combination of weight decay $(0.001, 0.004)$ and with learning rate of $0.001$. We also train the ResNet20 on CIFAR10 with one non-linear layer head and 1 to 6 linear layers on top. We use the same weight decay and learning rate. The relatively high variance of the results is due to averaging over rather strongly different weight decay values used per each depth. The plots clearly show that the NC2 significantly improves in the last layer as the depth of the linear head increases, while it gets slightly worse in the input layer to the linear head. This is consistent with our theory. The NC1 does not have a strong dependence with the number of layers.

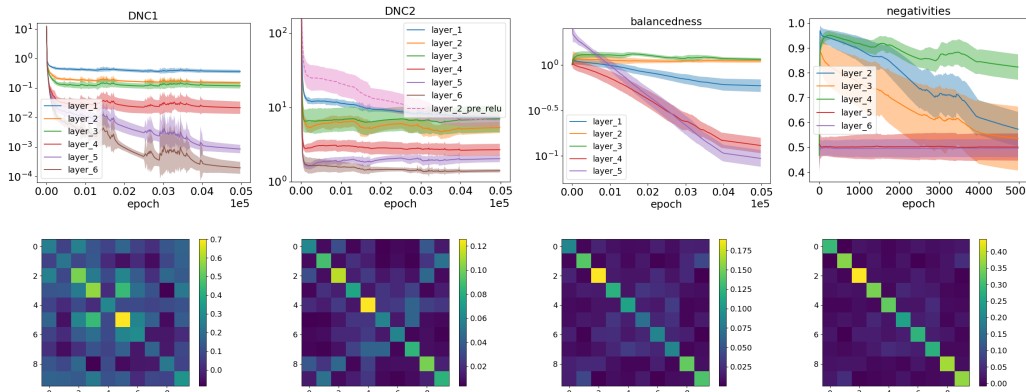

Figure 1: Last 7 layers of a 9-layer MLP trained on MNIST with weight decay 0.0018 and learning rate 0.001. **Top:** NC1s, NC2s, balancednesses and negativities, from left to right. Results are averaged over 5 runs, and the confidence band at 1 standard deviation is displayed. **Bottom:** Class-mean matrices of the last three layers (i.e., the linear head), the first before the last ReLU.

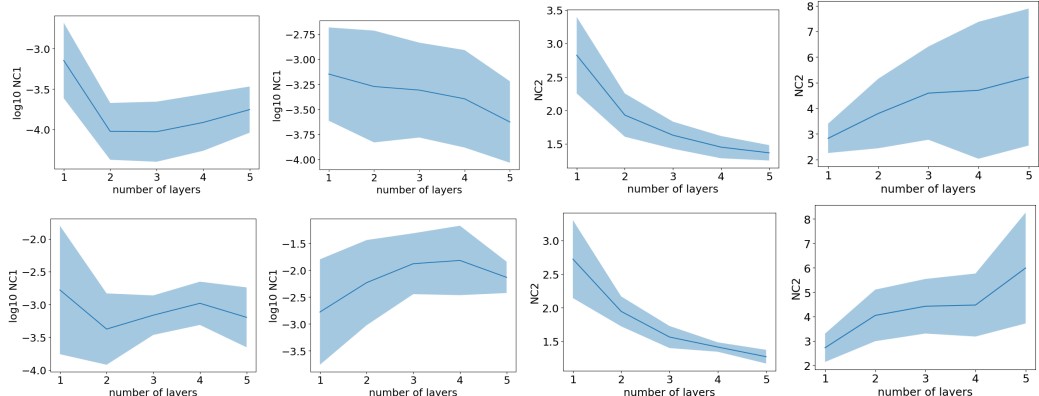

Figure 2: **Upper/Lower row:** MLP/ResNet20 with a deep linear head. **Left to right:** NC1 in the last layer; NC1 in the first layer of the linear head; NC2 in the last layer; NC2 in the first layer of the linear head. All plots are a function of the number of layers in the linear head. Results are averaged over 50 runs (5 runs for each of the 10 hyperparameter setups), and the confidence band at 1 standard deviation is displayed.

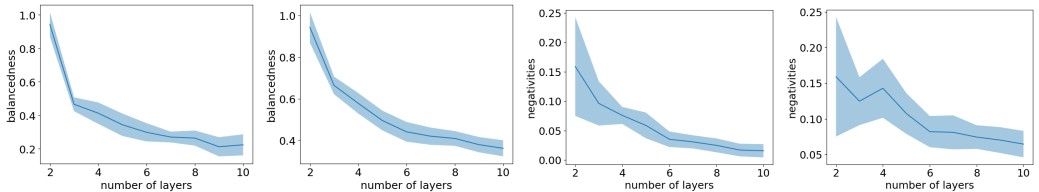

Figure 3: **Left to right:** Minimum balancedness; mean balancedness; minimum negativity; mean negativity across non-linear layers of the head as a function of the number of non-linear layers. Results are averaged over 10 runs (5 runs for each of the 2 hyperparameter setups), and the confidence band at 1 standard deviation is displayed.

**Linear layers are increasingly balanced throughout training.** Figure 1 shows that the metric capturing the balancedness exponentially decreases until it plateaus at a rather small value (due to the large learning rate) and then again it exponentially decreases at a smaller rate after reducing the learning rate. The balancedness of non-linear layers instead plateaus at a significantly larger value.

**Non-linear layers are increasingly balanced and linear, as the depth of the non-linear part increases.** In Figure 3, we show the dependence of balancedness and non-linearity of the non-

linear layers as a function of the depth of the non-linear part. We plot minimum balancedness and non-linearity across all layers, as well as mean balancedness and non-linearity. The depth of the non-linear part ranges from 4 to 12 (the first two layers are considered as backbone and not measured), the learning rate is either 0.001 or 0.002 (5 runs each), and the weight decay is 0.016 divided by the total number of layers. Balancedness clearly improves with depth, both on average per layer and in the most balanced layer. Similarly, the non-negativity of the layer that least uses the ReLU clearly decreases, and a decrease in the mean negativity is also reported (although less pronounced). Thus, these results suggest that the network tends to use non-linearities less as the depth increases and, in addition, it becomes more balanced. We also note that, in order to fit the data, regardless of the depth, the last non-linear layer(s) always exhibit significant negativity, heavily relying on the ReLU.

## 7 DISCUSSION AND CONCLUDING REMARKS

Our paper presents a framework to provably obtain neural collapse, which leads to the first result for the end-to-end training of deep networks. We now discuss our assumptions, compare with earlier work by Nguyen & Mondelli (2020) and with the NTK regime, concluding a future direction.

**Assumptions.** Theorem 3.1 provides a connection between neural collapse and properties of well-trained networks, i.e., approximate interpolation, approximate balancedness, bounded representations/weights and good conditioning of the features. As such, the result is rather general and requires no assumptions beyond the aforementioned properties. Theorem 4.4 then instantiates our framework for proving neural collapse to a class of networks with pyramidal topology (Assumption 4.1), smooth activations (Assumption 4.2) and for a class of initializations (Assumption 4.3). These assumptions are used only for the analysis of the first phase of the training dynamics, where the network achieves approximate interpolation. Thus, they could be replaced by any set of assumptions guaranteeing that gradient descent reaches small training loss. Specifically, such guarantees are obtained by Zou & Gu (2019) for deep ReLU networks (with stronger requirements on over-parameterization but no assumptions on the topology) and by Bombari et al. (2022) for networks with minimum over-parameterization (under a requirement on the topology milder than Assumption 4.1). As concerns Assumption 4.3 on the initialization, we discuss in page 5 a setting where it holds. In addition, by following the argument in Appendix C of Nguyen & Mondelli (2020), one readily obtains that Assumption 4.3 also holds for the widely used LeCun's initialization, i.e., $W_\ell^0$ has i.i.d. Gaussian entries with variance $1/n_{\ell-1}$ for all $\ell \in [L]$, as long as $n_1 = \Omega(N)$.

**Comparison with Nguyen & Mondelli (2020).** The condition $n_1 = \Omega(N)$ mentioned above provides a strict improvement upon $n_1 = \Omega(N^2)$ required in Section 3.2 of (Nguyen & Mondelli, 2020) for LeCun's initialization. This is due to Assumption 4.3 being weaker than the corresponding condition on initialization in Nguyen & Mondelli (2020), see Assumption 3.1 therein. We now elaborate on similarities and differences between the proof of Theorem 4.4 and the analysis in Nguyen & Mondelli (2020). The strategy to handle the first phase of the dynamics in Theorem 4.4 is similar: the weights of the network reach an arbitrarily small loss without leaving a suitable ball centered at initialization. However, the implementation of this strategy is significantly different and our approach relies on Proposition 4.5. More precisely, the improvement comes from upper bounding the gradient as in (26) in Appendix B, which uses again the PL inequality. In contrast, Nguyen & Mondelli (2020) use the loose bound in (18) of their work. We also note that the analysis of the second phase of the dynamics is entirely new, as balancedness was not needed in Nguyen & Mondelli (2020).

**NTK regime and beyond.** Our analysis of the training dynamics in Theorem 4.4 is split into two phases. While the first phase uses typical NTK tools, the second phase is governed by the effects of weight decay, leading to a fundamentally different behavior. In fact, at the end of the second phase, we observe low-rank weight matrices and feature learning, which could not appear with pure NTK dynamics. The two-phase behavior arises thanks to a separation of timescales as $\lambda$ gets smaller: the NTK phase lasts roughly $\Theta(\eta^{-1})$ steps, but the effect of weight decay is only observable over the longer timescale of the second phase, which lasts roughly $\Theta(\eta^{-1}\lambda^{-1})$ steps. This strategy gives us the best of both worlds: interpolation from the NTK regime, and balancedness from weight decay.

One important assumption in our work is that there is a linear head in the network, containing at least two layers. However, experimental evidence suggests that two linear layers are not necessary for collapse to happen. Thus, an exciting future direction is to use the approach developed here, e.g., the NTK analysis and the separation of timescales, to prove collapse without two linear layers.

## ACKNOWLEDGEMENTS

M. M. and P. S. are funded by the European Union (ERC, INF$^2$, project number 101161364). Views and opinions expressed are however those of the author(s) only and do not necessarily reflect those of the European Union or the European Research Council Executive Agency. Neither the European Union nor the granting authority can be held responsible for them.

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

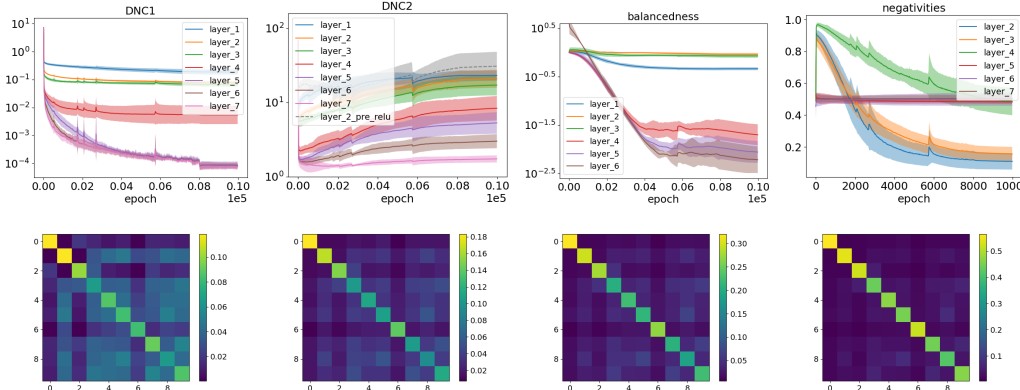

Figure 4: Last 7 layers of a 9-layer MLP trained on MNIST with weight decay $0.0018$ and learning rate $0.001$. **Top:** NC1s, NC2s, balancednesses and negativities, from left to right. Results are averaged over 5 runs, and the confidence band at 1 standard deviation is displayed. **Bottom:** Class-mean matrices of the last four layers (i.e., the linear head).

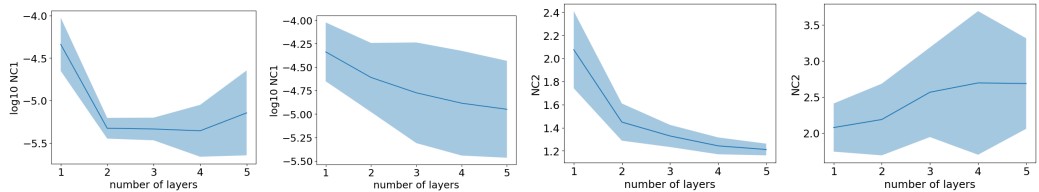

Figure 5: ResNet20 trained on MNIST with a deep linear head. **Left to right:** NC1 in the last layer; NC1 in the first layer of the linear head; NC2 in the last layer; NC2 in the first layer of the linear head. All plots are a function of the number of layers in the linear head. Results are based on 50 runs (5 runs for each of the 10 hyperparameter setups), and the confidence band at 1 standard deviation is displayed.

## A    ADDITIONAL EXPERIMENTS

We complement the experiments from Section 6 with additional numerical findings. We start by showing an analog of Figure 1 for MLP trained on MNIST to show that the behavior is robust with respect to the backbone and dataset. The results are shown in Figure 4. The architecture is an MLP with 5 non-linear layers followed by 4 linear layers. We use weight decay of $0.0018$ and learning rate of $0.001$, training for 10000 epochs. The plot fully agrees with the one in Figure 1 with ResNet20 training on CIFAR10 in every qualitative aspect, and it only differs in the numerical values attained by the NC metrics. Therefore, the conclusion is that the NC is attained across different architectures and NC2 progressively improves as we get closer to the last layer of the DNN.

Next, we extend Figure 2 with experiments on ResNet20 trained on MNIST, which are shown in Figure 5. As above, the results match the interpretations discussed in Section 6, which proves the robustness of our findings across different achitectures, datasets and hyperparameter settings.

## B    DEFERRED PROOFS

**Theorem B.1** (More precise version of Theorem 3.1). *If the network satisfies*

- *approximate interpolation, i.e., $\|Z_L - Y\|_F \leq \epsilon_1$,*

- *approximate balancedness, i.e., $\left\|W_{\ell+1}^\top W_{\ell+1} - W_\ell W_\ell^\top\right\|_{op} \leq \epsilon_2$, for $\ell \in \{L_1 + 1, \ldots, L - 1\}$,*

- *bounded representations and weights, i.e., $\|Z_{L-2}\|_{op}, \|Z_{L-1}\|_{op}, \|W_\ell\|_{op} \le r$, for $\ell \in \{L_1 + 1, \ldots, L\}$,*

*then if $\epsilon_1 \le \min\left(s_K(Y), \sqrt{\frac{(K-1)N}{4K}}\right)$,*

$$NC1(Z_{L-1}) \le \frac{r^2}{N} \frac{(\Psi(\epsilon_1, \epsilon_2, r))^2}{\left(\sqrt{\frac{K-1}{K}} - \frac{2}{\sqrt{N}}\epsilon_1\right)^2}, \tag{18}$$

*where $\Psi(\epsilon_1, \epsilon_2, r) = r\left(\frac{\epsilon_1}{s_K(Y)-\epsilon_1} + \sqrt{n_{L-1}\epsilon_2}\right)$. If we additionally assume that the linear part of the network is not too ill-conditioned, i.e., $\kappa(W_{L:L_1+1}) \le c_3$, then*

$$\kappa(W_L) \le c_3^{\frac{1}{L_2}}(1+\epsilon)^{\frac{1}{L_2}} + c_3^{\frac{1}{L_2}-1}\epsilon, \tag{19}$$

*with $\epsilon = \frac{\frac{L_2^2}{2}r^{2(L_2-1)}\epsilon_2}{\frac{(s_K(Y)-\epsilon_1)^2}{\|X\|_{op}^2 r^{2L_1}} - \frac{L_2^2}{2}r^{2(L_2-1)}\epsilon_2}$. Finally,*

$$NC2(Z_{L-1}) \le \frac{\kappa(W_L) + s_K(Y)^{-1}r\Psi(\epsilon_1, \epsilon_2, r)}{1 - s_K(Y)^{-1}r\Psi(\epsilon_1, \epsilon_2, r)} \tag{20}$$

$$NC3(Z_{L-1}, W_L) \ge \frac{(\sqrt{N}-\epsilon_1)^2 + N(\kappa(W_L))^{-2} - \left(r\Psi(\epsilon_1, \epsilon_2, r) + \sqrt{K}(\kappa(W_L)^2 - 1)\right)^2}{2N\kappa(W_L)(1+\epsilon_1)}. \tag{21}$$

By performing Taylor expansions and some algebraic manipulations, one readily obtains (2), (3), (4), (5) from (18), (19), (20), (21), respectively.

*Proof of Theorem B.1.* **NC1:** We start by proving the claim (18) on NC1. We have

$$s_K(Y) - \epsilon_1 \le s_K(Y + (Z_L - Y)) = s_K(Z_L) = s_K(W_L Z_{L-1}) \le s_K(W_L)r, \tag{22}$$

where the first inequality is the application of Weyl's inequality for singular values and the last one uses $s_k(A)s_1(B) \ge s_1(AB)$. Denote $W_L^+$ the pseudoinverse of $W_L$. Then, we have that $W_L^+ W_L$ equals the projection $P$ on the row space of $W_L$. We can now write

$$Z_{L-1} = PZ_{L-1} + (I-P)Z_{L-1} = W_L^+ Y + W_L^+(Z_L - Y) + (I-P)W_{L-1}Z_{L-2}.$$

Note that

$$\left\|W_L^+(Z_L - Y)\right\|_F \le \frac{\epsilon_1 r}{s_K(Y) - \epsilon_1},$$

since $s_K(W_L) \ge (s_K(Y) - \epsilon_1)/r$ by (22). Furthermore,

$$\|(I-P)W_{L-1}Z_{L-2}\|_F^2 \le \|(I-P)W_{L-1}\|_F^2 \|Z_{L-2}\|_{op}^2 \le r^2 \mathrm{tr}((I-P)W_{L-1}W_{L-1}^\top)$$
$$= r^2 \mathrm{tr}((I-P)(W_{L-1}W_{L-1}^\top - W_L^\top W_L)) \le r^2 n_{L-1}\epsilon_2.$$

Putting these together, we have:

$$\left\|Z_{L-1} - W_L^+ Y\right\|_F \le r\left(\frac{\epsilon_1}{s_K(Y) - \epsilon_1} + \sqrt{n_{L-1}\epsilon_2}\right) = \Psi(\epsilon_1, \epsilon_2, r). \tag{23}$$

From now on, we will drop the index $L-1$ and treat everything without layer label as belonging to that layer (the membership to any other layer will be indexed). We have that

$$\mathrm{tr}(\Sigma_W) = \mathrm{tr}\left(\frac{1}{N}\sum_{c,i}(z_{ci} - \mu_c)(z_{ci} - \mu_c)^\top\right) = \frac{1}{N}\sum_{c,i}\|z_{ci} - \mu_c\|_2^2$$
$$\le \frac{1}{N}\sum_{c,i}\left\|z_{ci} - (W_L^+)_{c:}\right\|_2^2 = \frac{1}{N}\left\|Z_{L-1} - W_L^+ Y\right\|_F^2.$$

Furthermore,

$$\text{tr}(\Sigma_B) = \text{tr}\left(\frac{1}{K}\sum_{c=1}^{K}(\mu_c - \mu_G)(\mu_c - \mu_G)^\top\right) = \frac{1}{K}\sum_{c=1}^{K}\|\mu_c - \mu_G\|_2^2$$

$$\geq \frac{\|W_L\|_{op}^2}{r^2}\frac{1}{K}\sum_{c=1}^{K}\|\mu_c - \mu_G\|_2^2 \geq \frac{1}{Kr^2}\sum_{c=1}^{K}\|\mu_c^L - \mu_G^L\|_2^2.$$

Now we proceed by lower-bounding the last term:

$$\frac{1}{K}\sum_{c=1}^{K}\|\mu_c^L - \mu_G^L\|_2^2 \geq \left(\frac{1}{K}\sum_{c=1}^{K}\|\mu_c^L - \mu_G^L\|_2\right)^2$$

$$\geq \left(\frac{1}{K}\sum_{c=1}^{K}\|\mu_c^Y - \mu_G^Y\|_2 - \frac{1}{K}\sum_{c=1}^{K}\|\mu_c^Y - \mu_c^L\|_2 - \|\mu_G^Y - \mu_G^L\|_2\right)^2,$$

where $\mu_c^Y, \mu_G^Y$ are the class and global means of the label matrix $Y$. A direct computation yields that $\frac{1}{K}\sum_{c=1}^{K}\|\mu_c^Y - \mu_G^Y\| = \sqrt{\frac{K-1}{K}}$. Next we have:

$$\|\mu_G^Y - \mu_G^L\|_2 = \left\|\frac{1}{N}\sum_{c,i}z_{ci}^L - \frac{1}{N}\sum_{c,i}z_{ci}^Y\right\|_2 \leq \frac{1}{N}\sum_{c,i}\|z_{ci}^L - z_{ci}^Y\|_2$$

$$\leq \sqrt{\frac{1}{N}\sum_{c,i}\|z_{ci}^L - z_{ci}^Y\|_2^2} = \frac{1}{\sqrt{N}}\|Z_L - Y\|_F \leq \frac{\epsilon_1}{\sqrt{N}}.$$

Finally, for any fixed $c$ we have

$$\|\mu_c^Y - \mu_c^L\|_2 = \left\|\frac{1}{n}\sum_{i}z_{ci}^Y - \frac{1}{n}\sum_{i}z_{ci}^L\right\|_2 \leq \frac{1}{n}\sum_{i}\|z_{ci}^Y - z_{ci}^L\|_2.$$

Therefore we get:

$$\frac{1}{K}\sum_{c=1}^{K}\|\mu_c^Y - \mu_c^L\|_2 \leq \frac{1}{N}\sum_{c,i}\|z_{ci}^Y - z_{ci}^L\|_2 \leq \frac{1}{\sqrt{N}}\|Z_L - Y\|_F \leq \frac{\epsilon_1}{\sqrt{N}}.$$

Upper bounding these terms in the above computation and dividing $\text{tr}(\Sigma_W)$ by $\text{tr}(\Sigma_B)$ yields (18).

**Conditioning of $W_L$:** We now prove the claim (19) on the conditioning of $W_L$. We rely first on Lemma C.2 to relate the singular values of $W_L$ to the $L_2$-th root of those of $W_{L:L_1+1}$. We therefore obtain that

$$\frac{s_1(W_L)^{2L_2}}{s_K(W_L)^{2L_2}} \leq \frac{s_1(W_{L:L_1+1})^2 + \frac{L_2^2}{2}r^{2(L_2-1)}\epsilon_2}{s_K(W_{L:L_1+1})^2 - \frac{L_2^2}{2}r^{2(L_2-1)}\epsilon_2}$$

$$= \frac{s_1(W_{L:L_1+1})^2}{s_K(W_{L:L_1+1})^2} + \frac{s_K(W_{L:L_1+1})^2\frac{L_2^2}{2}r^{2(L_2-1)}\epsilon_2 + s_1(W_{L:L_1+1})^2\frac{L_2^2}{2}r^{2(L_2-1)}\epsilon_2}{\left(s_K(W_{L:L_1+1})^2 - \frac{L_2^2}{2}r^{2(L_2-1)}\epsilon_2\right)s_K(W_{L:L_1+1})^2}.$$

We have that

$$s_K(Z_L) \leq s_K(W_{L:L_1+1})\|Z_{L_1}\|_{op} \leq s_K(W_{L:L_1+1})\|X\|_{op}r^{L_1},$$

and therefore

$$s_K(W_{L:L_1+1}) \geq \frac{s_K(Z_L)}{\|X\|_{op}r^{L_1}} \geq \frac{s_K(Y) - \epsilon_1}{\|X\|_{op}r^{L_1}}.$$

This gives that

$$\frac{s_1(W_L)^{2L_2}}{s_K(W_L)^{2L_2}} \leq \frac{s_1(W_{L:L_1+1})^2}{s_K(W_{L:L_1+1})^2} + \frac{1 + \kappa(W_{L:L_1+1})^2}{\left(\frac{(s_K(Y)-\epsilon_1)^2}{\|X\|_{op}^2r^{2L_1}} - \frac{L_2^2}{2}r^{2(L_2-1)}\epsilon_2\right)}\frac{L_2^2}{2}r^{2(L_2-1)}\epsilon_2.$$

Thus, the following chain of inequalities gives (19):

$$\kappa(W_L) \leq \left(\kappa(W_{L:L_1+1})^2(1+\epsilon)+\epsilon\right)^{\frac{1}{2L_2}}$$
$$\leq \kappa(W_{L:L_1+1})^{\frac{1}{L_2}}(1+\epsilon)^{\frac{1}{2L_2}} + \kappa(W_{L:L_1+1})^{\frac{1}{L_2}-1}(1+\epsilon)^{\frac{1}{2L_2}-1}\epsilon$$
$$\leq \kappa(W_{L:L_1+1})^{\frac{1}{L_2}}(1+\epsilon)^{\frac{1}{2L_2}} + \kappa(W_{L:L_1+1})^{\frac{1}{L_2}-1}\epsilon,$$

using the concavity of the $2L_2$-th root.

**NC(2+3):** We start with the proof of the claim (21) on NC3, as the derivations will be used in the proof of the claim (20) on NC2 later. Since the cosine similarity does not depend on the scale of the involved matrices, we perform the following rescaling: denoting $\alpha = \|W_L\|$, we define $W_L' = \frac{W_L}{\alpha}$ and $Z_{L-1}' = \alpha Z_{L-1}$. Similarly to before we denote $(z')_{ci}^{(L-1)}$ the $i$-th sample of the $c$-th class in the matrix $Z_{L-1}'$. Then, we write

$$\text{NC3}(Z_{L-1}', W_L') = \frac{1}{N}\sum_{c,i}\cos((z')_{ci}^{(L-1)}, (W_L')_{c:}) = \frac{1}{N}\sum_{c,i}\frac{\left\langle (z')_{ci}^{(L-1)}, (W_L')_{c:}\right\rangle}{\left\|(z')_{ci}^{(L-1)}\right\|_2 \|(W_L')_{c:}\|_2}$$

$$\geq \frac{\sum_{c,i}\left\langle (z')_{ci}^{(L-1)}, (W_L')_{c:}\right\rangle}{N\kappa(W_L)(1+\epsilon_1)} = \frac{2\left\langle Z_{L-1}', (W_L')^\top Y\right\rangle}{2N\kappa(W_L)(1+\epsilon_1)}$$

$$= \frac{\left\|Z_{L-1}'\right\|_F^2 + \left\|(W_L')^\top Y\right\|_F^2 - \left\|Z_{L-1}' - (W_L')^\top Y\right\|_F^2}{2N\kappa(W_L)(1+\epsilon_1)}.$$

Here, the first inequality follows from upper bounding $\|(W_L')_{c:}\|_2$ trivially by 1 and

$$\left\|(z')_{ci}^{(L-1)}\right\|_2 \leq \kappa(W_L)\left\|z_{ci}^{(L)}\right\|_2 \leq \kappa(W_L)\left(\left\|y_c - z_{ci}^{(L)}\right\|_2 + \|y_c\|_2\right) \leq \kappa(W_L)(1+\epsilon_1).$$

Now,

$$\left\|Z_{L-1}'\right\|_F^2 \geq \|Z_L\|_F^2 \geq (\|Y\|_F - \|Z_L - Y\|_F)^2 = (\sqrt{N}-\epsilon_1)^2.$$

Furthermore, we readily have that $\left\|(W_L')^\top Y\right\|_F \geq \frac{\sqrt{N}}{\kappa(W_L)}$. Finally, we have:

$$\left\|Z_{L-1}' - (W_L')^\top Y\right\|_F \leq \left\|Z_{L-1}' - (W_L')^+ Y\right\|_F + \left\|(W_L')^+ Y - (W_L')^\top Y\right\|_F.$$

From (23), we obtain that

$$\left\|Z_{L-1}' - (W_L')^+ Y\right\|_F \leq r^2\left(\frac{\epsilon_1}{s_K(Y)-\epsilon_1} + \sqrt{n_{L-1}\epsilon_2}\right) = r\Psi(\epsilon_1, \epsilon_2, r). \tag{24}$$

Finally, we proceed with upper bounding $\left\|(W_L')^+ Y - (W_L')^\top Y\right\|_F$, which can be done via a sandwich bound on the singular values using the conditioning number. This gives

$$\left\|(W_L')^+ Y - (W_L')^\top Y\right\|_F \leq \sqrt{K}\left(\kappa(W_L) - \frac{1}{\kappa(W_L)}\right) \leq \sqrt{K}(\kappa(W_L)^2 - 1).$$

Putting all the obtained bounds together, we get the desired bound (21) on NC3.

Finally, to pass from the bound (19) on $\kappa(W_L)$ to the bound (20) on NC2, we use the inequality in (24) obtained in the proof of NC3. Note that obtaining a bound on $\kappa(\bar{Z}_{L-1}')$ is equivalent to obtaining a bound on $\kappa(\bar{Z}_{L-1})$ since multiplying by a scalar does not change the condition number. Using (24) we get:

$$\left\|\bar{Z}_{L-1}' - (W_L')^+\right\|_F = \left\|Z_{L-1}'Y^+ - (W_L')^+ YY^+\right\|_F \leq \left\|Y^+\right\|_{op}\left\|Z_{L-1}' - (W_L')^+ Y\right\|_F$$

$$\leq \frac{r\Psi(\epsilon_1, \epsilon_2, r)}{s_K(Y)}.$$

As $\|(W_L')^+\|_{op} = \kappa(W_L)$ and $s_K((W_L')^+) = 1$, we conclude that

$$\kappa(\bar{Z}_{L-1}') \leq \frac{\kappa(W_L) + r\Psi(\epsilon_1, \epsilon_2, r)(s_K(Y))^{-1}}{1 - r\Psi(\epsilon_1, \epsilon_2, r)(s_K(Y))^{-1}},$$

which gives the desired bound in (20).

$\square$

**Proposition 4.5.** *Let $C_0(\theta)$ satisfy the $\alpha$-PL inequality* (10) *in the ball $B(\theta_0, r_0)$. Then, in the same ball, $C_\lambda(\theta)$ satisfies the inequality*

$$\|\nabla C_\lambda(\theta))\|_2^2 \geq \frac{\alpha}{4}\left(C_\lambda(\theta) - \lambda m_\lambda\right), \tag{11}$$

*where $m_\lambda = (1 + \sqrt{4\lambda/\alpha})^2 \left(\|\theta_0\|_2 + r_0\right)^2$. Furthermore, assume that $r_0 \geq 8\sqrt{C_\lambda(\theta_0)/\alpha}$ and $\nabla C_0(\theta)$ is $\beta_1$-Lipschitz in $B(\theta_0, r_0)$. Then, for any $\eta < 1/(2\beta_1)$, there exists*

$$k_1 \leq \left\lceil \frac{\log \frac{\lambda m_\lambda}{C_\lambda(\theta_0) - \lambda m_\lambda}}{\log(1 - \eta\frac{\alpha}{8})} \right\rceil \tag{12}$$

*such that the $k_1$-th iterate of GD satisfies*

$$C_\lambda(\theta_{k_1}) \leq 2\lambda m_\lambda, \qquad \|\theta_{k_1} - \theta_0\|_2 \leq 8\sqrt{\frac{C_\lambda(\theta_0)}{\alpha}} \leq r_0. \tag{13}$$

*Proof.* Let $\theta \in B(\theta_0, r_0)$. Then, the following chain of inequalities holds:

$$
\begin{aligned}
\|\nabla C_0(\theta) + \lambda\theta\|_2^2 &\geq (\|\nabla C_0(\theta)\|_2 - \lambda\|\theta\|_2)^2 \\
&\geq \left(\sqrt{\frac{\alpha}{2}C_0(\theta)} - \lambda\|\theta\|_2\right)^2 \\
&= \left(\sqrt{\frac{\alpha}{2}C_\lambda(\theta) - \frac{\alpha\lambda}{4}\|\theta\|_2^2} - \lambda\|\theta\|_2\right)^2 \\
&\geq \left(\sqrt{\frac{\alpha}{2}C_\lambda(\theta)} - \left(\lambda + \sqrt{\frac{\alpha\lambda}{4}}\right)\|\theta\|_2\right)^2 \\
&\geq \frac{\alpha}{4}C_\lambda(\theta) - \lambda\left(\sqrt{\frac{\alpha}{4}} + \sqrt{\lambda}\right)^2\|\theta\|_2^2 \\
&\geq \frac{\alpha}{4}\left(C_\lambda(\theta) - \lambda\left(1 + \sqrt{\frac{4\lambda}{\alpha}}\right)^2 (\|\theta_0\|_2 + r_0)^2\right).
\end{aligned} \tag{25}
$$

Here, in the second line we use that $\sqrt{\frac{\alpha}{2}C_0(\theta)} \geq \lambda\|\theta\|_2$, which follows from $C_\lambda(\theta) \geq \lambda m_\lambda$ (otherwise, the claim is trivial); in the fourth line we use that $\sqrt{a-b} \geq \sqrt{a} - \sqrt{b}$ for $a \geq b$; in the fifth line we use that $(a-b)^2 \geq \frac{a^2}{2} - b^2$ for all $a, b$; and in the sixth line we use that $\theta$ is in the ball $B(\theta_0, r_0)$. As the LHS of (25) equals $\|\nabla C_\lambda(\theta))\|_2^2$, this proves (11).

Next, let $(\theta_k)_{k\in\mathbb{N}}$ be the GD trajectory. Pick $k$ s.t. $\theta_k \in B(\theta_0, r_0)$. Then,

$$
\begin{aligned}
C_\lambda(\theta_{k+1}) - C_\lambda(\theta_k) &= -\eta\int_0^1 \langle\nabla C_\lambda(\theta_k - s\eta\nabla C_\lambda(\theta_k)), \nabla C_\lambda(\theta_k)\rangle ds \\
&= -\eta\|\nabla C_\lambda(\theta_k)\|_2^2 + \eta\int_0^1 \langle\nabla C_\lambda(\theta_k - s\eta\nabla C_\lambda(\theta_k)) - \nabla C_\lambda(\theta_k), \nabla C_\lambda(\theta_k)\rangle ds \\
&\leq -\eta\|\nabla C_\lambda(\theta_k)\|_2^2 + \eta\int_0^1 \|\nabla C_\lambda(\theta_k)\|_2 \|\nabla C_\lambda(\theta_k - s\eta\nabla C_\lambda(\theta_k)) - \nabla C_\lambda(\theta_k)\|_2\, ds \\
&\leq -\eta\|\nabla C_\lambda(\theta_k)\|_2^2 + \eta^2\beta_1\|\nabla C_\lambda(\theta_k)\|_2^2 \\
&= -\eta(1 - \eta\beta_1)\|\nabla C_\lambda(\theta_k)\|_2^2 \\
&\leq -\frac{\eta}{2}\|\nabla C_\lambda(\theta_k)\|_2^2 \\
&\leq -\eta\frac{\alpha}{8}\left(C_\lambda(\theta_k) - \lambda m_\lambda\right).
\end{aligned}
$$

Here, in the fourth line we use that the gradient is $\beta_1$-Lipschitz in $B(\theta_0, r_0)$; in the sixth line we use that $\eta < 1/(2\beta_1)$; and in the last line we use (11). Thus, as long as $\theta_j \in B(\theta_0, r_0)$ for all $j \in [k]$, by iterating the argument above, we have

$$C_\lambda(\theta_k) - \lambda m_\lambda \leq (C_\lambda(\theta_0) - \lambda m_\lambda)\left(1 - \eta\frac{\alpha}{8}\right)^k.$$

This readily implies that, by letting $k_1$ be the first index $k$ s.t. $C_\lambda(\theta_k) \le 2\lambda m_\lambda$, (12) holds. Finally, the distance $\|\theta_{k_1} - \theta_0\|_2$ is upper bounded by

$$
\begin{aligned}
\sum_{k=0}^{k_1-1} \eta \|\nabla C_\lambda(\theta_k)\|_2 &= \eta \sum_{k=0}^{k_1-1} \frac{\|\nabla C_\lambda(\theta_k)\|_2^2}{\|\nabla C_\lambda(\theta_k)\|_2} \\
&\le \frac{4}{\sqrt{\alpha}} \sum_{k=0}^{k_1-1} \frac{C_\lambda(\theta_k) - C_\lambda(\theta_{k+1})}{\sqrt{C_\lambda(\theta_k) - \lambda m_\lambda}} \\
&= \frac{4}{\sqrt{\alpha}} \sum_{k=0}^{k_1-2} \frac{C_\lambda(\theta_k) - C_\lambda(\theta_{k+1})}{\sqrt{C_\lambda(\theta_k) - \lambda m_\lambda}} + \frac{4}{\sqrt{\alpha}} \frac{C_\lambda(\theta_{k_1-1}) - C_\lambda(\theta_{k_1})}{\sqrt{C_\lambda(\theta_{k_1-1}) - \lambda m_\lambda}} \\
&\le \frac{8}{\sqrt{\alpha}} \sum_{k=0}^{k_1-2} \frac{C_\lambda(\theta_k) - C_\lambda(\theta_{k+1})}{\sqrt{C_\lambda(\theta_{k+1}) - \lambda m_\lambda} + \sqrt{C_\lambda(\theta_k) - \lambda m_\lambda}} + \frac{8}{\sqrt{\alpha}} \frac{C_\lambda(\theta_{k_1-1}) - \lambda m_\lambda}{\sqrt{C_\lambda(\theta_{k_1-1}) - \lambda m_\lambda}} \\
&= \frac{8}{\sqrt{\alpha}} \sum_{k=0}^{k_1-2} \left( \sqrt{C_\lambda(\theta_k) - \lambda m_\lambda} - \sqrt{C_\lambda(\theta_{k+1}) - \lambda m_\lambda} \right) + \frac{8}{\sqrt{\alpha}} \sqrt{C_\lambda(\theta_{k_1-1}) - \lambda m_\lambda} \\
&= \frac{8}{\sqrt{\alpha}} \sqrt{C_\lambda(\theta_0) - \lambda m_\lambda} \le \frac{8}{\sqrt{\alpha}} \sqrt{C_\lambda(\theta_0)},
\end{aligned}
$$
(26)

which concludes the proof. $\qquad\square$

**Theorem B.2** (More precise version of Theorem 4.4). *Let the network satisfy Assumption 4.1, $\sigma$ satisfy Assumption 4.2 and the initial conditions satisfy Assumption 4.3. Fix $0 < \epsilon_1 \le \frac{1}{2}\sqrt{\frac{(K-1)N}{K}}$, $\epsilon_2 > 0$, let $b \ge 1$ be s.t. $\|X_{:i}\|_2 \le b$ for all $i$, and run $k$ steps of $\lambda$-regularized GD with step size $\eta$, where*

$$
\lambda \le \min\left( 2\left(\frac{\gamma}{2}\right)^{L-2} \lambda_F \lambda_{3\to L}, \frac{2C_0(\theta_0)}{\|\theta_0\|_2^2}, \frac{\epsilon_1^2}{18(\|\theta_0\|_2 + \lambda_F/2)^2} \right),
$$

$$
\eta \le \min\left( \frac{1}{2\beta_1}, \frac{1}{5N\beta b^3 \max\left(1, \left(\frac{2\epsilon_1^2}{\lambda}\right)^{3L/2}\right) L^{5/2}}, \frac{1}{2\lambda}, \left(\frac{\lambda}{2\epsilon_1^2}\right)^{L_1+L} \frac{\epsilon_2}{4\|X\|_{op}^2} \right), \quad (27)
$$

$$
k \ge \left\lceil \frac{\log \frac{\lambda m_\lambda}{C_\lambda(\theta_0) - \lambda m_\lambda}}{\log(1 - \eta\frac{\alpha}{8})} \right\rceil + \left\lceil \frac{\log \frac{\lambda \epsilon_2}{4\epsilon_1^2}}{\log(1 - \eta\lambda)} \right\rceil,
$$

*with $\beta_1 = 5N\beta b^3 \left(\prod_{\ell=1}^L \max(1, \bar{\lambda}_\ell)\right)^3 L^{5/2}$, $m_\lambda = (1 + \sqrt{4\lambda/\alpha})^2 (\|\theta_0\|_2 + r_0)^2$, $r_0 = \frac{1}{2}\min(\lambda_F, \min_{\ell\in\{3,\dots,L\}} \lambda_\ell)$, and $\alpha = 2^{-(L-3)}\gamma^{L-2}\lambda_F\lambda_{3\to L}$. Then, we have that*

$$
NC1(Z_{L-1}^k) \le \frac{r^2}{N} \frac{(\Psi(\epsilon_1\sqrt{2}, \epsilon_2, r))^2}{\left(\sqrt{\frac{K-1}{K}} - \frac{2\sqrt{2}}{\sqrt{N}}\epsilon_1\right)^2}, \quad (28)
$$

*with $\Psi(\epsilon_1, \epsilon_2, r) = r\left(\frac{\epsilon_1}{s_K(Y) - \epsilon_1} + \sqrt{n_{L-1}\epsilon_2}\right)$ and*

$$
r = \max\left( \epsilon_1\sqrt{\frac{2}{\lambda}}, \left(\epsilon_1\sqrt{\frac{2}{\lambda}}\right)^{L-2} \|X\|_{op}, \left(\epsilon_1\sqrt{\frac{2}{\lambda}}\right)^{L-1} \|X\|_{op} \right). \quad (29)
$$

Since the terms $\beta_1, m_\lambda, r_0, \alpha$ do not depend on $\epsilon_1, \epsilon_2$ (but just on the network architecture and initialization), taking $\lambda = \Theta(\epsilon_1^2)$ and $\eta = O(\epsilon_2)$ (as in the statement of Theorem 4.4) satisfies the first two requirements in (27), also giving that $r$ in (29) is of constant order. Finally, as $\log(1-x) \approx -x$ for small $x$, the quantity $\eta k$ – which quantifies the time of the dynamics, since it is the product of learning rate and number of GD steps – is of order $\log(1/\lambda) + \log(1/\epsilon_2)/\lambda$. Thus, after some manipulations, one obtains that $k = \Omega\left(\frac{1}{\epsilon_1^2\epsilon_2} \log\frac{1}{\epsilon_2}\right)$ (as in the statement of Theorem 4.4).

*Proof of Theorem B.2.* By Lemma 4.1 in (Nguyen & Mondelli, 2020), the loss $C_0(\theta)$ satisfies the $\alpha$-PL inequality with $\alpha = 4\gamma^{L-2}\sigma_{\min}(Z_1)\prod_{p=3}^{L}\sigma_{\min}(W_p)$, where we have also used that $\sigma'$ is lower bounded by $\gamma$ by Assumption 4.2. Thus, by taking $r_0 = \frac{1}{2}\min(\lambda_F, \min_{\ell \in \{3,\dots,L\}}\lambda_\ell)$, we have that, for all $\theta \in B(\theta_0, r_0)$, $C_0(\theta)$ satisfies the $\alpha$-PL inequality with $\alpha = 2^{-(L-3)}\gamma^{L-2}\lambda_F\lambda_{3\to L}$.

By using Assumption 4.3 and that $\lambda$ is upper bounded by $\frac{2C_0(\theta_0)}{\|\theta_0\|_2^2}$ in (27), one can readily verify that $r_0 \geq 8\sqrt{C_\lambda(\theta_0)/\alpha}$. Furthermore, by Lemma C.1, we have that $\nabla C_0(\theta)$ is $\beta_1$-Lipschitz for all $\theta \in B(\theta_0, r_0)$. Hence, we can apply Proposition 4.5, which gives that, for some $k_1$ upper bounded in (12),
$$C_\lambda(\theta_{k_1}) \leq 2\lambda m_\lambda \leq \epsilon_1^2,$$
where the last inequality uses again the upper bound on $\lambda$ in (27).

With gradient flow, the regularized loss would only decrease further after $k_1$ steps. Since we are working with gradient descent, we simply need to assume that the learning rate is small enough to guarantee a decreasing loss. If the gradient $\nabla C_\lambda(\theta)$ is $\beta_2$-Lipschitz, then
$$
\begin{aligned}
C_\lambda(\theta_{k+1}) - C_\lambda(\theta_k) &= C_\lambda(\theta_k - \eta\nabla C_\lambda(\theta_k)) - C_\lambda(\theta_k) \\
&= -\eta\int_0^1 \langle \nabla C_\lambda(\theta_k - s\eta\nabla C_\lambda(\theta_k)), \nabla C_\lambda(\theta_k)\rangle \, ds \\
&\leq -\eta\|\nabla C_\lambda(\theta_k)\|_2^2 \\
&\quad + \eta\|\nabla C_\lambda(\theta_k)\|_2 \max_{s\in[0,1]}\|\nabla C_\lambda(\theta_k) - \nabla C_\lambda(\theta_k - s\eta\nabla C(\theta_k))\|_2 \\
&\leq -\eta\|\nabla C_\lambda(\theta_k)\|_2^2 + \eta^2\beta_2\|\nabla C_\lambda(\theta_k)\|_2^2 \\
&= -\eta(1 - \eta\beta_2)\|\nabla C_\lambda(\theta_k)\|_2^2,
\end{aligned}
$$
which is non-positive as long as $\eta \leq 1/\beta_2$. Furthermore, as long as the regularized loss is decreasing, the parameter norm is bounded by $\epsilon_1\sqrt{2/\lambda}$. Lemma C.1 then implies that the gradient is $\beta_2 = 5N\beta b^3\max\left(1, \epsilon_1^{3L}\left(\frac{2}{\lambda}\right)^{3L/2}\right)L^{5/2}$-Lipschitz and, therefore, $\eta \leq 1/\beta_2$ holds by (27). This allows us to conclude that, for all $k$ satisfying the lower bound in (27), $C_\lambda(\theta_k) \leq \epsilon_1^2$, hence the network achieves approximate interpolation, i.e.,
$$\|Z_L^k - Y\|_F \leq \epsilon_1\sqrt{2}. \tag{30}$$

Next, we show approximate balancedness. To do so, for $\ell \in \{L_1 + 2, \dots, L - 1\}$, we define
$$T_\ell^k := (W_{\ell+1}^k)^\top \cdots (W_L^k)^\top (Z_L^k - Y)(Z_{L_1}^k)^\top (W_{L_1+1}^k)^\top \cdots (W_{\ell-1}^k)^\top.$$
Then, we have
$$
\begin{aligned}
W_\ell^{k+1}(W_\ell^{k+1})^\top &= ((1-\eta\lambda)W_\ell^k - \eta T_\ell^k)((1-\eta\lambda)W_\ell^k - \eta T_\ell^k)^\top \\
&= (1-\eta\lambda)^2 W_\ell^k(W_\ell^k)^\top - (1-\eta\lambda)\eta(W_\ell^k(T_\ell^k)^\top + T_\ell^k(W_\ell^k)^\top) + \eta^2 T_\ell^k(T_\ell^k)^\top.
\end{aligned}
$$
Similarly,
$$
\begin{aligned}
&(W_{\ell+1}^{k+1})^\top W_{\ell+1}^{k+1} \\
&= (1-\eta\lambda)^2(W_{\ell+1}^k)^\top W_{\ell+1}^k - (1-\eta\lambda)\eta((W_{\ell+1}^k)^\top T_{\ell+1}^k + (T_{\ell+1}^k)^\top W_{\ell+1}^k) + \eta^2(T_{\ell+1}^k)^\top T_{\ell+1}^k.
\end{aligned}
$$
Let us define
$$D_\ell^k = (W_{\ell+1}^k)^\top W_{\ell+1}^k - W_\ell^k(W_\ell^k)^\top.$$
Since $T_\ell^k(W_\ell^k)^\top = (W_{\ell+1}^k)^\top T_{\ell+1}^k$ and $W_\ell^k(T_\ell^k)^\top = (T_{\ell+1}^k)^\top W_{\ell+1}^k$, we have
$$D_\ell^{k+1} = (1-\eta\lambda)^2 D_\ell^k + \eta^2((T_{\ell+1}^k)^\top T_{\ell+1}^k - T_\ell^k(T_\ell^k)^\top).$$
Recall that, for all $k$ lower bounded in (27), $\|Z_L^k - Y\|_F \leq \epsilon_1\sqrt{2}$ and $\|W_\ell^k\|_F \leq \|\theta^k\|_2 \leq \epsilon_1\sqrt{2/\lambda}$, which also implies that $\|Z_{L_1}^k\|_F \leq \left(\epsilon_1\sqrt{2/\lambda}\right)^{L_1}\|X\|_{op}$. Thus,
$$
\begin{aligned}
\|D_\ell^{k+1}\|_{op} &\leq (1-\eta\lambda)^2\|D_\ell^k\|_{op} + \eta^2(\|T_\ell^k\|_{op}^2 + \|T_{\ell+1}^k\|_{op}^2) \\
&\leq (1-\eta\lambda)^2\|D_\ell^k\|_{op} + \eta^2\|Z_{L_1}^k\|_{op}^2\|Z_L^k - Y\|_{op}^2\left(\prod_{j\neq\ell}\|W_j^k\|_{op}^2 + \prod_{j\neq\ell+1}\|W_j^k\|_{op}^2\right) \\
&\leq (1-\eta\lambda)^2\|D_\ell^k\|_{op} + 4\eta^2\epsilon_1^2\left(\frac{2\epsilon_1^2}{\lambda}\right)^{L_1+L-1}\|X\|_{op}^2.
\end{aligned}
$$

By using the upper bounds $\eta \leq \frac{1}{2\lambda}$ and $\eta \leq \left(\frac{\lambda}{2\epsilon_1^2}\right)^{L_1+L} \frac{\epsilon_2}{4\|X\|_{op}^2}$ in (27), we have that:

- if $\|D_\ell^k\|_{op} \geq \epsilon_2$, then $\|D_\ell^{k+1}\|_{op} \leq (1 - \eta\lambda)\|D_\ell^k\|_{op}$;

- if $\|D_\ell^k\|_{op} \leq \epsilon_2$, then $\|D_\ell^{k+1}\|_{op} \leq (1 - \eta\lambda)\epsilon_2 \leq \epsilon_2$.

This implies that, for all $\bar{k} \geq 0$ and $k_1 \geq \left\lceil \frac{\log \frac{\lambda m_\lambda}{C_\lambda(\theta_0) - \lambda m_\lambda}}{\log(1 - \eta\frac{\alpha}{8})} \right\rceil$,

$$\|D_\ell^{k_1 + \bar{k}}\|_{op} \leq \max((1 - \eta\lambda)^{\bar{k}}\|D_\ell^{k_1}\|_{op}, \epsilon_2).$$

Note that

$$\|D_\ell^{k_1}\|_{op} \leq \|W_{\ell+1}^{k_1}\|_{op}^2 + \|W_\ell^{k_1}\|_{op}^2 \leq \frac{4\epsilon_1^2}{\lambda},$$

which allows us to conclude that, for all $k$ lower bounded in (27),

$$\|D_\ell^k\|_2 \leq \epsilon_2.$$

Finally, we have the following bounds on the representations and weights at step $k$:

$$\left\|Z_{L-2}^k\right\|_{op} \leq \left(\epsilon_1\sqrt{\frac{2}{\lambda}}\right)^{L-2} \|X\|_{op},$$

$$\left\|Z_{L-1}^k\right\|_{op} \leq \left(\epsilon_1\sqrt{\frac{2}{\lambda}}\right)^{L-1} \|X\|_{op},$$

$$\left\|W_\ell^k\right\|_{op} \leq \epsilon_1\sqrt{\frac{2}{\lambda}}, \quad \text{for } \ell \in \{L_1 + 1, \dots, L\}.$$

Hence, an application of Theorem 3.1 proves the desired result. $\qquad \square$

**Proposition 5.1.** *Let $\sigma$ satisfy Assumption 4.2. Then, for any network that satisfies*

- *approximate interpolation, i.e., $\|Z_L - Y\|_F \leq \epsilon_1$,*

- *bounded parameters, i.e., $\|\theta\|_2^2 \leq LK + c$,*

*the linear part $W_{L:L_1+1}$ satisfies*

$$\kappa(W_{L:L_1+1}) \leq \exp\left(\frac{1}{2}\left(c + L_1 K \log K - 2K \log \frac{s_K(Y) - \epsilon_1}{\|X\|_{op}}\right)\right). \tag{14}$$

*Proof.* Let us split the parameter norm into the contribution from the nonlinear and linear layers:

$$\|\theta\|_2^2 = \|\theta_{nonlin}\|_2^2 + \|\theta_{lin}\|_2^2.$$

We first lower bound both parts in terms of the product of the linear part $W_{L:L_1+1}$. This then allows us to bound the conditioning of $W_{L:L_1+1}$. We start with the nonlinear part:

$$\|Z_{L_1}\|_F \leq \|X\|_{op} \prod_{\ell=1}^{L_1} \|W_\ell\|_F \leq \|X\|_{op} \left(\frac{1}{L_1}\|\theta_{nonlin}\|_2^2\right)^{\frac{L_1}{2}},$$

which implies that

$$\|\theta_{nonlin}\|_2^2 \geq L_1 \left(\frac{\|Z_{L_1}\|_F}{\|X\|_{op}}\right)^{\frac{2}{L_1}} \geq L_1 \left(\frac{\|Z_{L_1}\|_{op}}{\|X\|_{op}}\right)^{\frac{2}{L_1}}.$$

Since $\|Y - W_{L:L_1+1}Z_{L_1}\|_F \leq \epsilon_1$, we have

$$s_K(Y) \leq s_K(W_{L:L_1+1}Z_{L_1}) + \epsilon_1 \leq s_K(W_{L:L_1+1})\|Z_{L_1}\|_{op} + \epsilon_1,$$

so that $\|Z_{L_1}\|_{op} \geq \frac{s_K(Y) - \epsilon_1}{s_K(W_{L:L_1+1})}$ and thus

$$\|\theta_{nonlin}\|_2^2 \geq L_1 \left( \frac{s_K(Y) - \epsilon_1}{s_K(W_{L:L_1+1}) \|X\|_{op}} \right)^{\frac{2}{L_1}}. \tag{31}$$

Next, for the linear part, we know from Theorem 1 of Dai et al. (2021) that

$$\min_{(W_\ell)_{\ell=L_1+1}^L : A = W_{L:L_1+1}} \sum_{\ell=L_1+1}^L \|W_\ell\|_F^2 = (L - L_1) \sum_{i=1}^{\text{Rank}(A)} s_i(A)^{\frac{2}{L-L_1}},$$

so that

$$\|\theta_{lin}\|_2^2 \geq (L - L_1) \sum_{i=1}^{\text{Rank}(W_{L:L_1+1})} s_i(W_{L:L_1+1})^{\frac{2}{L-L_1}} \geq (L - L_1)K + 2 \log |W_{L:L_1+1}|_+, \tag{32}$$

where we used the fact that $x^{\frac{2}{L-L_1}} \geq 1 + \frac{2}{L-L_1} \log x$.

We also have following bound on the condition number $\kappa(W_{L:L_1+1})$:

$$\kappa(W_{L:L_1+1}) = \frac{s_1(W_{L:L_1+1})}{s_K(W_{L:L_1+1})} \leq \prod_{i=1}^K \frac{s_i(W_{L:L_1+1})}{s_K(W_{L:L_1+1})} = \frac{|W_{L:L_1+1}|_+}{s_K(W_{L:L_1+1})^K}. \tag{33}$$

By combining (31), (32), and (33) (after applying the log on both sides of (33)), we obtain

$$\|\theta\|_2^2 \geq L_1 \left( \frac{s_K(Y) - \epsilon_1}{s_K(W_{L:L_1+1}) \|X\|_{op}} \right)^{\frac{2}{L_1}} + (L - L_1)K + 2 \log |W_{L:L_1+1}|_+$$

$$\geq L_1 \left( \frac{s_K(Y) - \epsilon_1}{s_K(W_{L:L_1+1}) \|X\|_{op}} \right)^{\frac{2}{L_1}} + (L - L_1)K + 2 \log \kappa(W_{L:L_1+1}) + 2K \log s_K(W_{L:L_1+1}).$$

The above is lower bounded by the minimum over all possible choices of $s_K(W_{L:L_1+1})$. This minimum would be attained at $s_K(W_{L:L_1+1}) = K^{-\frac{L_1}{2}} \frac{s_K(Y) - \epsilon_1}{\|X\|_{op}}$, thus leading to the lower bound

$$\|\theta\|_2^2 \geq LK + 2 \log \kappa(W_{L:L_1+1}) - L_1 K \log K + 2K \log \frac{s_K(Y) - \epsilon_1}{\|X\|_{op}}.$$

This implies

$$2 \log \kappa(W_{L:L_1+1}) \leq \|\theta\|_2^2 - LK + L_1 K \log K - 2K \log \frac{s_K(Y) - \epsilon_1}{\|X\|_{op}}$$

$$\leq c + L_1 K \log K - 2K \log \frac{s_K(Y) - \epsilon_1}{\|X\|_{op}}.$$

$\square$

**Theorem 5.2.** *Let $\sigma$ satisfy Assumption 4.2. Assume there exist parameters of the nonlinear part $\theta_{nonlin} = (W_\ell)_{\ell=1}^{L_1}$ such that $Z_{L_1} = Y$ and $\|\theta_{nonlin}\|_2^2 = c$. Then, at any global minimizer of the regularized loss $\mathcal{L}_\lambda(\theta) = \frac{1}{2} \|Y - Z_L\|_F^2 + \frac{\lambda}{2} \|\theta\|_2^2$ with $\lambda \leq \frac{\epsilon_1^2}{KL+c}$, we have*

$$\kappa(W_{L:L_1+1}) \leq \left( \frac{\|X\|_{op}}{s_K(Y) - \epsilon_1} \right)^K \exp \left( \frac{1}{2} (c - L_1 K + L_1 K \log K) \right),$$

$$\kappa(W_L) \leq \left( \frac{\|X\|_{op}}{s_K(Y) - \epsilon_1} \right)^{\frac{K}{L_2}} \exp \left( \frac{1}{2L_2} (c - L_1 K + L_1 K \log K) \right). \tag{15}$$

*This implies that the bounds on NC1, NC2 and NC3 in (2), (4) and (5), respectively, hold with $\kappa(W_L)$ upper bounded as above and $\epsilon_2 = 0$.*

*Proof.* By assumption, there are parameters of the nonlinear part $\theta_{nonlin}$ such that the representation at the end of the nonlinear layers already matches the outputs $Z_{L_1} = Y$ with finite parameter norm $\|\theta_{nonlin}\|_2^2 = c$. We can now build a deeper network by simply setting all the linear layers to be the $K$-dimensional identity, so that $\|\theta_{lin}\|_2^2 = KL_2$, leading to a total parameter norm of $\|\theta\|_2^2 = c + KL_2$. Since the outputs matches the labels exactly, we have that the regularized loss is bounded by $\lambda(KL_2 + c)/2$. This implies that any global minimizer $\theta^*$ satisfies

$$\|Y - Z_L\|_F^2 \leq \lambda(KL_2 + c),$$
$$\|\theta^*\|_2^2 \leq KL_2 + c.$$

Choosing $\lambda \leq \frac{\epsilon_1^2}{KL_2+c}$, we obtain that the global minimizer $\theta^*$ must satisfy the assumptions of Proposition 5.1, which readily gives the desired upper bound on $\kappa(W_{L:L_1+1})$.

To finish the proof, we show that all critical points of the regularized loss have balanced linear layers, so that we may relate the conditioning of $W_L$ to that of the product $W_{L:L_1+1}$. At any critical point, the gradient w.r.t. to any weight matrix amongst the linear layers must be zero, that is

$$0 = \nabla_{W_\ell} C_\lambda(\theta) = W_{L:\ell+1}^\top (Z_L - Y) Z_{L_1}^\top W_{\ell-1:L_1+1}^\top + \lambda W_\ell.$$

This implies that $W_\ell = -\frac{1}{\lambda} W_{L:\ell+1}^\top (Z_L - Y) Z_{L_1}^\top W_{\ell-1:L_1+1}^\top$ for all linear layers $\ell$. Balancedness then follows directly

$$W_\ell W_\ell^\top = -\frac{1}{\lambda} W_{L:\ell+1}^\top (Z_L - Y) Z_{L_1}^\top W_{\ell:L_1+1}^\top = W_{\ell+1}^\top W_{\ell+1}.$$

Finally, the balancedness implies that

$$\kappa(W_L) = \kappa(W_{L:L_1+1})^{\frac{1}{L_2}} \leq \left( \frac{\|X\|_{op}}{s_K(Y) - \epsilon_1} \right)^{\frac{K}{L_2}} \exp\left( \frac{1}{2L_2} \left( c - L_1 K + L_1 K \log K \right) \right).$$

$\square$

**Proposition 5.3.** *For any network that satisfies*

- *bounded NTK, i.e.,* $\|\Theta\|_{op} = \max_A \frac{\|\nabla_\theta \mathrm{Tr}[Z_L A^T]\|_2^2}{\|A\|_F^2} \leq CL_2$,

- *approximate interpolation, i.e.,* $\|Z_L - Y\|_F \leq \epsilon_1$,

- *bounded weights, i.e.,* $\|W_\ell\|_{op} \leq r$,

*for any $M \leq L_2$, there is $\ell \in \{L_1 + 1, \ldots, L_1 + M\}$ such that $\kappa(W_{L:\ell}) \leq \frac{\sqrt{CL_2} K r}{\sqrt{M}(s_K(Y) - \epsilon_1)}$.*

*Furthermore, any network that satisfies approximate interpolation and bounded weights is such that*

$$\|\Theta\|_{op} \geq \frac{(s_K(Y) - \epsilon_1)^2}{K^2 r^2} L_2. \tag{17}$$

*Proof.* As the NTK is a sum over all layers, we can lower bound it by the contribution of the linear layers only. Formally, we have

$$\left\| \nabla_\theta \mathrm{Tr}[Z_L A^\top] \right\|_2^2 \geq \left\| \nabla_{\theta_{lin}} \mathrm{Tr}[Z_L A^\top] \right\|_2^2$$
$$= \sum_{\ell=L_1+1}^{L} \left\| W_{\ell-1:L_1+1} Z_{L_1} A^\top W_{L:\ell+1} \right\|_F^2$$

Let $v_1, \ldots, v_K \in \mathbb{R}^N$ be $K$ orthonormal vectors that span the preimage of $Z_L$ ($Z_L$ is rank $K$ as long as $\epsilon_1 \leq s_K(Y)$), and $e_1, \ldots, e_K \in \mathbb{R}^K$ be the standard basis of $\mathbb{R}^K$. We can then sum over $K^2$ possible choices of matrices $A = e_i v_j^T$ to obtain

$$K^2 \|\Theta\|_{op} \geq \sum_{i,j=1}^{K} \left\| \nabla_\theta (e_i^T Z_L v_j) \right\|_2^2 \geq \sum_{i,j=1}^{K} \sum_{\ell=L_1+1}^{L} \left\| W_{\ell-1:L_1+1} Z_{L_1} v_j \right\|_2^2 \left\| e_i^T W_{L:\ell+1} \right\|_2^2.$$

We know use the fact that $P_{\mathrm{Im}W_{L:\ell}^T}W_{\ell-1:L_1+1}Z_{L_1} = W_{L:\ell}^+W_{L:\ell}W_{\ell-1:L_1+1}Z_{L_1} = W_{L:\ell}^+Z_L$ to obtain the lower bound

$$K^2\|\Theta\|_{op} \geq \sum_{i,j=1}^{K}\sum_{\ell=L_1+1}^{L}\left\|W_{L:\ell}^+Z_Lv_j\right\|_2^2\left\|e_i^TW_{L:\ell+1}\right\|_2^2$$

$$= \sum_{\ell=L_1+1}^{L}\left\|W_{L:\ell}^+Z_L\right\|_F^2\left\|W_{L:\ell+1}\right\|_F^2$$

$$\geq s_K(Z_L)^2\sum_{\ell=L_1+1}^{L}\left\|W_{L:\ell}^+\right\|_F^2\frac{\|W_{L:\ell}\|_F^2}{\|W_\ell\|_{op}}$$

$$\geq \frac{(s_K(Y)-\epsilon_1)^2}{r^2}\sum_{\ell=L_1+1}^{L}\kappa(W_{L:\ell})^2.$$

This then implies that

$$CL_2 \geq \|\Theta\|_{op} \geq \frac{(s_K(Y)-\epsilon_1)^2}{K^2r^2}\sum_{\ell=L_1+1}^{L}\kappa(W_{L:\ell})^2.$$

Let us now assume by contradiction that for all the layers $\ell \in \{L_1+1, \ldots, L_1+M\}$ we have $\kappa(W_{L:\ell})^2 > \frac{CL_2K^2r^2}{M(s_K(Y)-\epsilon_1)^2}$, then

$$\frac{(s_K(Y)-\epsilon_1)^2}{K^2r^2}\sum_{\ell=L_1+1}^{L}\kappa(W_{L:\ell})^2 > CL_2,$$

which yields a contradiction. Therefore, there must be a layer within the layers $\ell \in \{L_1+1, \ldots, L_1+M\}$ such that $\kappa(W_{L:\ell})^2 \leq \frac{CL_2K^2r^2}{M(s_K(Y)-\epsilon_1)^2}$.

Furthermore, since $\kappa(W_{L:\ell}) \geq 1$, we know that

$$\|\Theta\|_{op} \geq \frac{(s_K(Y)-\epsilon_1)^2}{K^2r^2}L_2.$$

$\square$

## C   TECHNICAL RESULTS

**Lemma C.1.** *Let $b \geq 1$ be s.t. $\|X_{:i}\|_2 \leq b$ for all $i$. Then, inside the set of parameters with bounded weights $\|W_\ell\|_{op} \leq r_\ell$ for all $\ell \in [L]$ and $r_\ell \geq 1$, the gradient of the loss $\nabla C_0(\theta)$ is $5N\beta b^3\left(\prod_{j=1}^{L}r_j\right)^3 L^{5/2}$-Lipschitz.*

*Proof.* Consider two parameters $\theta = (W_\ell)_{\ell=1}^L, \theta' = (W_\ell')_{\ell=1}^L$, and let $Z_m, Z_m'$ be the corresponding outputs at layer $m$. Then, we obtain the following telescopic sum:

$$(Z_m)_{:i} - (Z_m')_{:i} = \sum_{\ell=1}^{L}(W_m\cdots W_{\ell+1}\circ\sigma\circ W_\ell\circ\sigma\circ W_{\ell-1}'\cdots\circ W_1')(X_{:i})$$
$$- (W_m\cdots W_{\ell+1}\circ\sigma\circ W_\ell'\circ\sigma\circ W_{\ell-1}'\circ\cdots\circ W_1')(X_{:i}), \tag{34}$$

so that

$$\|(Z_m)_{i:} - (Z_m')_{i:}\|_2 \leq \sum_{\ell=1}^{m}\prod_{j=1}^{m-1}r_j\|W_\ell - W_\ell'\|_F\|X_{:i}\|_2$$
$$\leq b\prod_{j=1}^{L-1}r_j\sqrt{m}\sqrt{\sum_{\ell=1}^{m}\|W_\ell - W_\ell'\|_F^2} \leq b\prod_{j=1}^{L-1}r_j\sqrt{m}\|\theta-\theta'\|_2. \tag{35}$$

Now, the gradient equals

$$\nabla C_0(\theta) = \left( \sum_{i=1}^N \prod_{j=1}^{\ell-1} D_{j,i} W_j X_{:i} (Y_{:i} - (Z_L)_{:i})^\top \prod_{j=\ell+1}^L W_j D_{j,i} \right)^\top_{\ell=1,\dots,L}$$

where $D_{j,i}$ is a diagonal matrix with diagonal entries given equal to the vector $\sigma'((Z_j)_{:i})$ for $j \in [L_1]$ and it is equal to the identity otherwise. Similarly, we define $D'_{j,i}$ as a diagonal matrix with diagonal entries given equal to the vector $\sigma'((Z'_j)_{:i})$ for $j \in [L_1]$ and equal to the identity otherwise. Since $\sigma'$ is $\beta$-Lipschitz, we have

$$\left\| D_{j,i} - D'_{j,i} \right\|_F \le \beta b \prod_{j=1}^{L-1} r_j \sqrt{m} \left\| \theta - \theta' \right\|_2 .$$

Furthermore since $\sigma' \le 1$, we have that $\left\| D_{j,i} \right\|_{op} \le 1$. By summing over all terms that need to be changed, we obtain

$$\|\nabla C_0(\theta) - \nabla C(\theta')\|_2 \le \sum_{\ell=1}^L \sum_{i=1}^N \|X_{:i}\|_2 \|Y_{:i} - (Z_L)_{:i}\|_2 \sum_{m \ne \ell} \prod_{j \notin \{m,L\}} r_j \|W_m - W'_m\|_F$$

$$+ \sum_{\ell=1}^L \sum_{i=1}^N \|X_{:i}\|_2 \|Y_{:i} - (Z_L)_{:i}\|_2 \sum_{m=1}^{L_1} \left( \prod_{j=1}^{L-1} r_j \right)^2 \beta \sqrt{m} \|\theta - \theta'\|_2 \|X_{:i}\|_2$$

$$+ \sum_{\ell=1}^L \sum_{i=1}^N \|X_{:i}\|_2 \left( \prod_{j=1}^{L-1} r_j \right)^2 \sqrt{L} \|\theta - \theta'\|_2 \|X_{:i}\|_2 ,$$

where the three terms correspond to the effect of changing the $W_\ell$'s, the $D_\ell$'s and the $Z_\ell$'s respectively. This can then be simplified to

$$\|\nabla C_0(\theta) - \nabla C_0(\theta)\|_2 \le N b^2 \left( 1 + \prod_{j=1}^L r_j \right) \prod_{j=1}^{L-1} r_j L^{3/2} \|\theta - \theta'\|_2$$

$$+ N b^3 \left( 1 + \prod_{j=1}^L r_j \right) \left( \prod_{j=1}^{L-1} r_j \right)^2 L L_1^{3/2} \beta \|\theta - \theta'\|_2$$

$$+ N b^2 \left( \prod_{j=1}^{L-1} r_j \right)^2 L^{3/2} \|\theta - \theta'\|_2$$

$$\le 5 N \beta b^3 \left( \prod_{j=1}^L r_j \right)^3 L^{5/2} \|\theta - \theta'\|_2 ,$$

where we use the fact that $\|(Z_L)_{:i}\|_2 \le b \prod_{j=1}^L r_j$. $\qquad\square$

**Lemma C.2.** *If the network satisfies*

- *approximate balancedness $\left\| W_{\ell+1}^T W_{\ell+1} - W_\ell W_\ell^T \right\|_{op} \le \epsilon_2$ for $\ell \in \{L_1 + 1, \cdots L - 1\}$,*

- *bounded weights $\|W_\ell\|_{op} \le r$ for $\ell \in \{L_1 + 1, \cdots, L\}$,*

*then we have $\left\| (W_L W_L^\top)^{L_2} - W_{L:L_1+1} W_{L:L_1+1}^\top \right\|_{op} \le \frac{L_2^2}{2} \epsilon_2 r^{2(L_2-1)}$, where $L_2 = L - L_1$.*

*Proof.* We denote $D_\ell := W_\ell W_\ell^\top - W_{\ell+1}^\top W_{\ell+1}$ and

$$E_\ell = \sum_{i=L_1+1}^\ell \left( W_{\ell+1}^\top W_{\ell+1} \right)^{i-L_1-1} D_\ell \left( W_\ell W_\ell^\top \right)^{\ell-i} .$$

We have that

$$
\begin{aligned}
\left(W_\ell W_\ell^\top\right)^{\ell-L_1} &= D_\ell \left(W_\ell W_\ell^\top\right)^{\ell-L_1-1} + W_{\ell+1}^\top W_{\ell+1} \left(W_\ell W_\ell^\top\right)^{\ell-L_1-1} \\
&= D_\ell \left(W_\ell W_\ell^\top\right)^{\ell-L_1-1} + W_{\ell+1}^\top W_{\ell+1} D_\ell \left(W_\ell W_\ell^\top\right)^{\ell-L_1-2} + \left(W_{\ell+1}^\top W_{\ell+1}\right)^2 \left(W_\ell W_\ell^\top\right)^{\ell-L_1-2} \\
&= \dots \\
&= \left(W_{\ell+1}^\top W_{\ell+1}\right)^{\ell-L_1} + \sum_{i=L_1+1}^{\ell} \left(W_{\ell+1}^\top W_{\ell+1}\right)^{i-L_1-1} D_\ell \left(W_\ell W_\ell^\top\right)^{\ell-i} \\
&= \left(W_{\ell+1}^\top W_{\ell+1}\right)^{\ell-L_1} + E_\ell.
\end{aligned}
$$

Then,

$$
\begin{aligned}
W_{L:L_1+1} W_{L:L_1+1}^\top &= W_L \cdots W_{L_1+1} W_{L_1+1}^\top \cdots W_L^\top \\
&= W_L \cdots W_{L_1+2} \left(W_{L_1+2}^\top W_{L_1+2} + E_{L_1+1}\right) W_{L_1+2}^\top \cdots W_L^\top \\
&= W_L \cdots W_{L_1+3} \left(W_{L_1+2}^\top W_{L_1+2}\right)^2 W_{L_1+3}^\top \cdots W_L^\top + \mathcal{E}_{L_1+1} \\
&= \dots \\
&= \left(W_L W_L^\top\right)^{L_2} + \sum_{i=L_1+1}^{L-1} \mathcal{E}_i,
\end{aligned}
$$

where $\mathcal{E}_i = W_L \cdots W_{i+1} E_i W_{i+1}^\top \cdots W_L^\top$. As a result, we have

$$
\left\| \left(W_L W_L^\top\right)^{L_2} - W_{L:L_1+1} W_{L:L_1+1}^\top \right\|_{op} = \left\| \sum_{i=L_1+1}^{L-1} \mathcal{E}_i \right\|_{op} \leq \sum_{i=L_1+1}^{L-1} \|\mathcal{E}_i\|_{op}.
$$

Since $\|W_k\|_{op} \leq r$ for all $k = \{L_1 + 1, \cdots L\}$, we have $\|E_\ell\|_{op} \leq (\ell - L_1)\epsilon_2 r^{2(\ell-L_1-1)}$ and $\|\mathcal{E}_\ell\|_{op} \leq r^{2(L-\ell)}(\ell - L_1)\epsilon_2 r^{2(\ell-L_1-1)} = (\ell - L_1)\epsilon_2 r^{2(L_2-1)}$. Thus, we have

$$
\left\| \left(W_L W_L^\top\right)^{L_2} - W_{L:L_1+1} W_{L:L_1+1}^\top \right\|_{op} \leq \frac{L_2^2}{2}\epsilon_2 r^{2(L_2-1)}.
$$

$\square$

