# OpenReview forum: "Wide Neural Networks Trained with Weight Decay Provably Exhibit Neural Collapse"
_ICLR.cc/2025/Conference — ICLR 2025 Oral_

### Official Review · Reviewer_39ne · 2024-10-27

**Soundness:** 3
**Presentation:** 3
**Contribution:** 3
**Rating:** 8
**Confidence:** 3

**Summary:**

This paper studies neural collapse phenomenon in training wide neural networks. The first result in this work establishes some general conditions (interpolation, balancedness between layers, well-conditioned-ness of weight matrices) such that NC1, NC2, and NC3 can hold. The second result considers training a wide neural network with gradient descent such that the aforementioned conditions hold after training, which implies neural collapse can happen after training.

**Strengths:**

This work goes beyond the conventional unconstrained feature model (which many previous works have worked on) and proves that neural collapse can happen after training for full-connected neural networks satisfying some architectural conditions such as pyramidal topology and smooth activation. The conditions established in Theorem 3.1 of this work under which neural collapse can happen seem very reasonable. Indeed, later, the authors proved that those conditions can be satisfied by training a neural network via gradient descent which automatically implies that neural collapse can happen after training.

I am not very familiar with the prior works along this line of research such as (Kothapalli & Tirer, 2024), (Hong & Ling, 2024) and many others mentioned in the introduction and related work. Based on my knowledge on neural collapse, I think this work made some interesting contributions towards understanding this phenomenon.

**Weaknesses:**

1. The analysis critically relies on the fact that the last two layers of the neural network are linear. I can definitely see this condition makes the problem a lot easier to analyze. I am wondering how hard it is to remove such restrictions.
2. It seems the analysis of Theorem 4.4 relies on the neural network in the NTK regime, as the pyramidal topology assumption has appeared in previous works such as (Nguyen & Mondelli, 2020). I don't regard this as a major weakness even if it turned out to be true that the networks are in the NTK regime given the contribution of this work, however, I do appreciate clarification on this.

**Questions:**

I am wondering whether the layer balanced-ness property after training can be proved as a direct consequence of [1] in the author's setting.

[1] Du, Simon S., Wei Hu, and Jason D. Lee. "Algorithmic regularization in learning deep homogeneous models: Layers are automatically balanced." Advances in neural information processing systems 31 (2018).

**Details Of Ethics Concerns:**

None.

---

> ### Author Response · Authors · 2024-11-18
> **Response**
>
> We thank the reviewer for recognizing the strengths of our work and for the detailed comments. We address each of them below.
>
> *W1: The analysis critically relies on the fact that the last two layers of the neural network are linear. I can definitely see this condition makes the problem a lot easier to analyze. I am wondering how hard it is to remove such restrictions.*
>
> **Response:** This is a great point. One important assumption in our work is that there is a linear head in the network, containing at least two layers. However, experimental evidence suggests that two linear layers are not necessary for collapse to happen. Removing this restriction is likely to be difficult and it provides an exciting future direction, which would consist of using the approach developed here, i.e., the NTK analysis and the separation of timescales, to prove collapse without two linear layers.
>
> *W2: It seems the analysis of Theorem 4.4 relies on the neural network in the NTK regime, as the pyramidal topology assumption has appeared in previous works such as (Nguyen & Mondelli, 2020). I don't regard this as a major weakness even if it turned out to be true that the networks are in the NTK regime given the contribution of this work, however, I do appreciate clarification on this.*
>
> **Response:** Well, yes and no. In the first phase, we are essentially in the NTK regime and use NTK-type techniques to guarantee convergence (technically the pyramidal network setup is not exactly the typical NTK regime, see the last paragraph of Section 1 in (Nguyen & Mondelli, 2020); however, this distinction does not play a major role in our analysis). Nevertheless, there is a second phase, where the weight decay starts to take effect which is not NTK-like. Though our control of this second dynamics is weaker, we can still guarantee that the network will become more balanced and remain interpolating. To be more precise, we have a separation of timescales as $\lambda$ gets smaller: the number of steps needed to reach balancedness is of order $\frac{1}{\eta \lambda}$ (weight-decay timescale). This is significantly later than the interpolation time which is of order $\frac{1}{\eta}$ (NTK timescale). Note that at the end of training we end up with low-rank weight matrices and feature learning (hidden features that are different from their initializations), which could not happen in a purely NTK regime, so the second phase plays an important role.
>
> This strategy/dynamics of “NTK followed by weight decay” is to our knowledge novel and it represents an important theoretical contribution of our paper. We have added a paragraph (**NTK regime and beyond**) in Section 7 of the revision about this aspect.
>
>
> *Q: I am wondering whether the layer balanced-ness property after training can be proved as a direct consequence of [1] in the author's setting.
> [1] Du, Simon S., Wei Hu, and Jason D. Lee. "Algorithmic regularization in learning deep homogeneous models: Layers are automatically balanced." Advances in neural information processing systems 31 (2018).*
>
> **Response:** This result is indeed very closely related, though what our analysis requires does not seem to be a direct consequence, mainly because we consider gradient descent instead of gradient flow, and we use weight decay/L2 regularization. Nevertheless, we have added a citation of this very relevant work.

---

> > ### Comment · Reviewer_39ne · 2024-11-26
> >
> > Thank you for your response. I am convinced by the contribution of this submission. I have raised my score to acceptance.

---

### Official Review · Reviewer_ggM6 · 2024-11-01

**Soundness:** 4
**Presentation:** 4
**Contribution:** 4
**Rating:** 8
**Confidence:** 4

**Summary:**

The authors identify a set of conditions under which various neural collapse phenomena provably occur in deep neural nets.  They consider neural nets which have a sequence of nonlinear layers and then a sequence of linear layers.  They find that approximate interpolation, weight balanced-ness, and boundedness suffice for deriving various neural collapse phenomena.  They then show that GD on networks with a "pyramidal" overparameterized topology (i.e., first width is >= number samples, remaining widths are decreasing), under suitable initialization and regularization, allow for one of the neural collapse phenomena to hold.  They then identify conditions (near-interpolation and small-norm) which ensure that global minimizers of the loss can satisfy all of the neural collapse phenomena.   Finally, they look at the neural collapse phenomena  from the perspective of the edge of stability via an analysis of the properties of the hessian under EoS assumptions.

**Strengths:**

The authors provide a novel characterization of conditions under which neural collapse can provably occur.  I am not an expert on the NC theory literature, but to my knowledge, no prior work identified balancedness and interpolation as being key factors which can allow for this to occur, and this is a pretty strong finding.   Since the balancedness of post-fixed linear layers is the strongest condition (as most common neural nets do not have multiple linear layers at the end, only a single one), the findings in Section 5 about how boundedness and interpolation can suffice for NC2+NC3 are also a nice addition.  The numerical findings nicely complement their theoretical ones.

**Weaknesses:**

There aren't any serious weaknesses to me.  The strongest assumptions--namely the need for multiple linear layers in order for the NC phenomenon to occur via the main theorem--seem necessary per experiments in Figure 1.  So it seems that these assumptions are strong for a fundamental reason.

The pyramidal structure assumption is a bit odd/strong, but only seems needed for the optimization result, which I don't think is the central point of the paper.

The authors don't provide a conclusion or discussion section, and it would have been useful to have comments from the authors about what they think are the weakest parts of their work and what work should be prioritized in the future.  I think they could get some additional space by removing some of the details about optimization from Sec 4 since I don't think they're super important.

**Questions:**

1. Are there no assumptions on the activation function $\sigma$ needed for Theorem 3.1?  None are stated in Section 3, although assumptino 4.2 appears later.  It seems odd that a potentially discontinuous and horribly behaved activation function would be permitted (I imagine Lipschitz is required?)

2. Assumption 4.2 seems more like a smooth leaky relu, which has appeared in prior work [Chatterjee arXiv:2203.16462, Frei et al COLT 2022]

3. The authors talk about (NC1)-(NC3), what about (NC4) from the original Papyan paper?

4. Are there any important differences between the proof of Theorem 4.4 and the proofs/results in Nguyen and Mondelli?   I'm not familiar with that work, but it seems like it's not super necessary to have details about optimization (e.g. the PL inequality) in the main section of the paper, it distracts from what I think are the more interesting and stronger results elswhere in it (it would also give additional space to have a proper conclusion and outline what the authors think are future directions).   I am assuming balancedness didn't appear in the prior analysis but does here.  A few sentences describing high-level differences between proofs and ideas would be helpful.

5. Also, to be clear, the optimization analysis in Sec 4 is in the NTK regime right?  I didn't see this explicitly mentioned but it should be if it is known.

---

> ### Author Response · Authors · 2024-11-18
> **Response (1)**
>
> We thank the reviewer for the positive evaluation of our work and for the insightful comments. We address each of them below.
>
> *W: The authors don't provide a conclusion or discussion section, and it would have been useful to have comments from the authors about what they think are the weakest parts of their work and what work should be prioritized in the future. I think they could get some additional space by removing some of the details about optimization from Sec 4 since I don't think they're super important.*
>
> **Response:** This is a great point. Following the reviewer’s suggestion, in the revision we have added a final section titled ‘Discussion and Concluding Remarks’. There, we discuss our assumptions, we compare with earlier work by Nguyen & Mondelli (2020) (see also the response to Q4 below), we discuss how our analysis relates to the NTK regime (see also the response to Q5 below), and we conclude with a direction for future research.
>
> To make space and for the sake of readability, we have simplified the statements of Theorems 3.1 and 4.4 using a big-O notation, as suggested by Reviewer 1hwn.
>
>
> *Q1: Are there no assumptions on the activation function  needed for Theorem 3.1? None are stated in Section 3, although assumptino 4.2 appears later. It seems odd that a potentially discontinuous and horribly behaved activation function would be permitted (I imagine Lipschitz is required?)*
>
> **Response:** Theorem 3.1 concerns the linear head containing at least two layers, after the non-linear part of the network. For this reason, no additional assumption is needed on the activation function.
>
> *Q2: Assumption 4.2 seems more like a smooth leaky relu, which has appeared in prior work [Chatterjee arXiv:2203.16462, Frei et al COLT 2022]*
>
> **Response:** This is a good point, thank you for pointing out these works. Our assumption is in fact the same as in [Frei et al, COLT 2022] and it is similar to that made in [Chatterjee arXiv:2203.16462]. We have edited the revision accordingly.
>
> *Q3: The authors talk about (NC1)-(NC3), what about (NC4) from the original Papyan paper?*
>
> **Response:** This is another good point. In the original Papyan paper (as well as in follow-ups), it has been shown that the NC4 property follows from the first three. Thus, if one proves the first three, the fourth one is automatically guaranteed. We did not discuss this explicitly, since many neural collapse papers already addressed this [4] and NC4 is not usually discussed in the recent NC papers [1, 2, 3].
>
> [1] Tirer, Tom, Haoxiang Huang, and Jonathan Niles-Weed. "Perturbation analysis of neural collapse." International Conference on Machine Learning. PMLR, 2023.
>
> [2] Súkeník, Peter, Marco Mondelli, and Christoph H. Lampert. "Deep neural collapse is provably optimal for the deep unconstrained features model." Advances in Neural Information Processing Systems 36 (2023).
>
> [3] Jiang, Jiachen, et al. "Generalized neural collapse for a large number of classes." arXiv preprint arXiv:2310.05351 (2023).
>
> [4] Wu, Robert, and Vardan Papyan. "Linguistic Collapse: Neural Collapse in (Large) Language Models." arXiv preprint arXiv:2405.17767 (2024).

---

> > ### Author Response · Authors · 2024-11-18
> > **Response (2)**
> >
> > *Q4: Are there any important differences between the proof of Theorem 4.4 and the proofs/results in Nguyen and Mondelli? I'm not familiar with that work, but it seems like it's not super necessary to have details about optimization (e.g. the PL inequality) in the main section of the paper, it distracts from what I think are the more interesting and stronger results elsewhere in it (it would also give additional space to have a proper conclusion and outline what the authors think are future directions). I am assuming balancedness didn't appear in the prior analysis but does here. A few sentences describing high-level differences between proofs and ideas would be helpful.*
> >
> > **Response:** We now address this point in detail in the paragraph **Comparison with Nguyen & Mondelli (2020)** added in Section 7 of the revision. There, we elaborate on similarities and differences between the proof of Theorem 4.4 and the analysis in Nguyen & Mondelli (2020).
> >
> > The strategy to handle the first phase of the dynamics in Theorem 4.4 is similar to that in Nguyen & Mondelli (2020): the weights of the network reach an arbitrarily small loss without leaving a suitable ball centered at initialization. However, the implementation of this strategy is significantly different and our approach relies on Proposition 4.5. More precisely, the improvement comes from upper bounding the gradient as in (26) in Appendix B, which uses the PL inequality. In contrast, Nguyen & Mondelli (2020) use the loose bound in (18) of their work. We also note that the analysis of the second phase of the dynamics is entirely new. In fact, the purpose of this second phase is to achieve balancedness, which was not needed by Nguyen & Mondelli (2020) as mentioned by the reviewer. We elaborate on the novelty of our proof strategy also in the subsequent paragraph of Section 7 **NTK regime and beyond**, see the response to Q5 below.
> >
> > Given that we do provide an improvement upon the earlier analysis by Nguyen & Mondelli (2020) via the PL inequality of Proposition 4.5, we have opted to keep it in the revision. To make space, we have instead simplified the statements of Theorem 3.1 and 4.4. By doing so, we have also been able to discuss a future direction, as suggested by the reviewer, see the concluding paragraph of Section 7 of the revision.
> >
> > *Q5: Also, to be clear, the optimization analysis in Sec 4 is in the NTK regime right? I didn't see this explicitly mentioned but it should be if it is known.*
> >
> > **Response:** Well, yes and no. In the first phase, we are essentially in the NTK regime and use NTK-type techniques to guarantee convergence (technically the pyramidal network setup is not exactly the typical NTK regime, see the last paragraph of Section 1 in (Nguyen & Mondelli, 2020); however, this distinction does not play a major role in our analysis). Nevertheless, there is a second phase, where the weight decay starts to take effect which is not NTK-like. Though our control of this second dynamics is weaker, we can still guarantee that the network will become more balanced and remain interpolating. To be more precise, we have a separation of timescales as $\lambda$ gets smaller: the number of steps needed to reach balancedness is of order $\frac{1}{\eta \lambda}$ (weight-decay timescale). This is significantly later than the interpolation time which is of order $\frac{1}{\eta}$ (NTK timescale). Note that at the end of training we end up with low-rank weight matrices and feature learning (hidden features that are different from their initializations), which could not happen in a purely NTK regime, so the second phase plays an important role.
> >
> > This strategy/dynamics of “NTK followed by weight decay” is to our knowledge novel and it represents an important theoretical contribution of our paper. We have added a paragraph (**NTK regime and beyond**) in Section 7 of the revision about this aspect.

---

> > > ### Comment · Reviewer_ggM6 · 2024-11-29
> > >
> > > Thanks for the detailed response.  I think this is a strong paper and it is improved with the described revisions.

---

### Official Review · Reviewer_1hwn · 2024-11-02

**Soundness:** 3
**Presentation:** 1
**Contribution:** 2
**Rating:** 8
**Confidence:** 3

**Summary:**

This paper provides theoretical evidence for end-to-end training of true deep neural networks. This is contrary to previous works which primarily rely on the unconstrained features model. The paper provides explicit bounds on NC1, NC2 and NC3 for both globally optimal ($\ell_2$) regularized deep neural networks as well as neural networks trained with gradient descent. The results provide new insights on the role of additional linear layers, weight decay regularization and large learning rates with respect to neural collapse.

**Strengths:**

- The authors provide the first theoretical analysis of neural collapse that does not rely on the unconstrained features model.
- The use of additional linear layers is novel and interesting technique. To the best of my knowledge such an architecture has not been studied.
- The results apply to deep neural networks trained with gradient descent (under a particular architecture/nonstandard activation function) as well as networks which are globally optimal for the weight decay regularized objective.

**Weaknesses:**

- My main concern with the paper is the writing. The theoretical statements are very difficult to parse this is especially the case for eqs 2-5 in Theorem 3.1. Can the authors provide a more interpretable version of this theorem (hide constants, big-O etc.) and delay the details to the appendix?
- I have the same concern about Theorem 4.4.

### Minor
- The proof of Theorem 3.1 in the Appendix could also be more clear. For example mentioning Weyl's inequality in the first inequality of (21)
- In the proof of Theorem 5.2 at the Appendix (line 1264) shouldn't it be
$$\kappa(W_L) = \kappa(W_{L:L_1+1})^{\frac{1}{L_2}}$$
not
$$\kappa(W_L) = \kappa(W_{L:L_1+1})^{\frac{1}{L_1}}$$
- Same concern about $L_1$ vs $L_2$ in the statement of Theorem 5.2.

**Questions:**

- What exactly is the role of these additional linear layers?  Are they only required for proving NC2 and NC3?
- Do these results suggest that neural networks with 1  non-layer and many linear layers can also exhibit neural collapse?

---

> ### Author Response · Authors · 2024-11-18
> **Response**
>
> We thank the reviewer for appreciating our work and for the detailed comments. We reply to each of the points raised in the review below. We have also uploaded a revised version of the manuscript highlighting in red the main changes.
>
> *W1: My main concern with the paper is the writing. The theoretical statements are very difficult to parse this is especially the case for eqs 2-5 in Theorem 3.1. Can the authors provide a more interpretable version of this theorem (hide constants, big-O etc.) and delay the details to the appendix?*
>
> **Response:** This is a very good point. Following the suggestion of the reviewer, we have re-written Theorem 3.1 in a more interpretable form hiding constants and using a big-O notation. The precise version of this result is now deferred to Appendix B (cf. Theorem B.1).
>
> *W2: I have the same concern about Theorem 4.4.*
>
> **Response:** Similarly, we have simplified the statement of Theorem 4.4, deferring the original version to Appendix B (cf. Theorem B.2).
>
> *W3 (Minor): The proof of Theorem 3.1 in the Appendix could also be more clear. For example mentioning Weyl's inequality in the first inequality of (21).*
>
> **Response:** Thank you for pointing this out, we have made the intermediate steps of the proof more clear in the revision.
>
> *W4 (Minor): In the proof of Theorem 5.2 at the Appendix (line 1264) shouldn’t it be $\kappa(W_L)=\kappa(W_{L:L_1+1})^{\frac{1}{L_2}}$ not $\kappa(W_L)=\kappa(W_{L:L_1+1})^{\frac{1}{L_1}}$*
>
> **Response:** Thanks for noticing this typo. We have corrected it in the revision.
>
> *W5 (Minor): Same concern about $L_1$ vs $L_2$ in the statement of Theorem 5.2.*
>
> **Response:** Thanks for noticing this typo. We have corrected it in the revision.
>
> *Q1: What exactly is the role of these additional linear layers? Are they only required for proving NC2 and NC3?*
>
> **Response:** We also need them for our proof of NC1, although there we only require two consecutive linear layers. In particular we need the balancedness property to ensure that the features $Z_{L-1}$ are approximately in the row space of $W_L$ so that the formula $Z_{L-1}=W_L^+Z_L$ approximately holds.
>
>
> *Q2: Do these results suggest that neural networks with 1 non-layer and many linear layers can also exhibit neural collapse?*
>
> **Response:** Yes, our results imply that neural networks with 1 non-linear layer and many linear layers provably exhibit neural collapse.

---

### Official Review · Reviewer_PRAd · 2024-11-04

**Soundness:** 4
**Presentation:** 3
**Contribution:** 4
**Rating:** 8
**Confidence:** 2

**Summary:**

This paper presents an interesting theoretical advancement in understanding neural collapse in practical, end-to-end training scenarios, moving beyond the unconstrained features model. The authors provide a rigorous demonstration that neural collapse arises in networks with linear layers appended to a nonlinear backbone, given conditions of interpolation and balancedness. They show that these conditions hold for sufficiently wide networks trained with gradient descent and L2 regularization. The empirical results further support the theoretical findings, showcasing the robustness of neural collapse across various architectures and datasets.

**Strengths:**

This work provides an interesting theoretical insight on the role of training algorithm in the emergence of neural collapse, which I found especially exciting, and I think it opens up new directions for understanding the generalization properties of deep learning models.

**Weaknesses:**

In my opinion, this is a solid paper and I can not think of a weakness.

**Questions:**

In Figure 3 the authors' numerical results show that non-linear layers are increasingly linear, as the depth of the non-linear part increases. Could the authors provide more insights into how this observation relates to their theoretical results and the mechanisms driving this increased linearity?

---

> ### Author Response · Authors · 2024-11-18
> **Response**
>
> We thank the reviewer for appreciating our work. We answer the question below.
>
> *Q: In Figure 3 the authors' numerical results show that non-linear layers are increasingly linear, as the depth of the non-linear part increases. Could the authors provide more insights into how this observation relates to their theoretical results and the mechanisms driving this increased linearity?*
>
> **Response:** This experimental observation serves as an empirical justification for the usage of linear layers at the end of the network. The more linear the non-linear layers are, the better justified is our usage of linear layers not only as a theoretical construction, but also as a phenomenon that is reasonable in practice.
>
> As for the mechanisms driving this, our intuition is that once the network extracts all the relevant features of the training data, it is best for it to just carry these features all the way to the end of the network. This seems to minimize the total $\ell_2$-norm of the weight matrices. For more intuition on this topic please also see [1], where the author analyzes this emergence of linearity in detail.
>
> [1] Jacot, Arthur. "Bottleneck structure in learned features: Low-dimension vs regularity tradeoff." Advances in Neural Information Processing Systems 36 (2023): 23607-23629.

---

> > ### Comment · Reviewer_PRAd · 2024-11-30
> >
> > Thank you to the author for their thoughtful response and the insightful reference. I think it is a strong paper that brings a nice perspective on the theory of neural collapse.

---

### Official Review · Reviewer_H9Md · 2024-11-04

**Soundness:** 3
**Presentation:** 2
**Contribution:** 3
**Rating:** 6
**Confidence:** 4

**Summary:**

Dear authors, thank you for submitting your work to ICLR 2025. The paper considers the phenomenon of 'neural collapse' and attempt to extend current rather special case results to brader class of networks, specifically, a deep neural (non-linear) networks with a wide first layer, funnel architecture and several, i.e. 2+, linear layers (head) before the output. After taking several assumptions, paper shows in series of Theorems (3.1, 4.4,5.2) and Propositions (4.5, 5.1,5.3) that GD training with weight decay (under further assumptions) leads to within class variability collapse (NC1). Results are supported by experiments on MNIST and CIFAR and MLP and ResNet + MLP head.

**Strengths:**

S1: Novelty, pushing for (more) general results on timely and attractive topic of 'neural collapse' in DNN training
S2: Striving for a theoretically solid argumentation stating necessary assumptions (in Theorems and Propositions) and as Assumtions 4.1, 4.2, 4.3 ...
S3: Well done Introduction positioning paper within existing works (UFM, NTK and other specific results)

**Weaknesses:**

W1: Accessibility of the paper for a '15 mins' reader. The main results are formulated using (Abstract, l19) 'balancedness', (Abstract, l20) 'bounded conditioning' and other only later defined phrases and makes it hard to asses the attractivity/usefullness of paper unless reading it whole. It is recommended to rework (the Abstract at least, if no Conclusions are present) to make it clear and self-sustained.

W2: Quite a few assumptions are required in general. More over thay are added along the way, e.g., Assumption 4.2, $|\sigma(x)| \leq |x|$, etc., (which is ok, but also narrows the applicability of the final result). Some of those are very technical (especialy those in Theorems, such as in Theorem 3.1, (2),(3), (4)) but as well Assumption 4.3, Theorem 4.4. (10) and more) and with paper specific notation to learn, e.g., $\lambda_{3 \rightarrow L}$. It would help paper to have an thorough discussion on applicability/limitations of these assumptions. Perhaps at expense of shortening some 'proof sketches' refering them to SM?

W3: 'Discussion and Conclusion' sections are not presented. This is most likely due to space constrained and will be added in case of acceptance (?). Yet, I find it impacts the paper quality negatively. Especially in a light of the previous (W2) comments, the discussion and conclusions could have brought a critical 'back to reality' summary. Idealy it would bring more intuition for results and their application. Yet, I find it very important to have such Discussion reviewed before publishing ...

W4: Some references are rather inaccurately interpreted/generalized too much perhaps. For instance the lines 399-400 "...Thankfully, instead of simply diverging, for large η (but not too large) the parameters naturally end up at the ‘edge of stability’: the top eigenvalue of the Hessian is close to $2/\eta$ , i.e., the threshold below which GD is stable..." from (Cohen et all. 2021). Referenced work provides only experimental evidence for a phonomenon and only approximately, i.e., for certain settings operator norm of Hessian exceeds 'stability threshold' $2/\eta$, etc. Than approximation used on l410, especially $O(\epsilon_1)$ is only valid if $\nabla^2_{\theta} Z_L$ norm is bounded, which is ok for NTK regime, but not necessarily for large learning rate regime. Or is it?

W5: Following up on W4, Proposition 5.3, and other Theorem combine NTK with large learning rate regime, which sounds dangerous. Also requirement on wide first layer, suggest a NTK limit case is required. Could authors clarify a bit more on this relation?

Overall, I find it to be a solid attractive paper with technically legit reasoning, taking few shortcuts (some noted above) and with missing discussion and conclusions. I suggest authors to work on alleviating weaknesses and discussing limitations to improve contributions of this interesting work significantly.

**Questions:**

See Weaknesses for the most concerning questions.
Additionaly:

Q1: Line 188: "... aproaches 1 ..." Shouldn't it be "... aproaches 2" based on (3)? Is it still sufficient for orthogonality claims (can the proof be adjusted to account for it)?

Q2: Proof sketch of Theorem 4.4., lines "306". Why and how is are "two phases" guaranteed to happen during GD training?

---

> ### Author Response · Authors · 2024-11-18
> **Response (1)**
>
> We thank the reviewer for appreciating the novelty of our work and our theoretical analysis, as well as for the detailed comments. We reply to each of the points raised in the review below. We have also uploaded a revised version of the manuscript highlighting in red the main changes.
>
> *W1: Accessibility of the paper for a '15 mins' reader. The main results are formulated using (Abstract, l19) 'balancedness', (Abstract, l20) 'bounded conditioning' and other only later defined phrases and makes it hard to assess the attractivity/usefullness of paper unless reading it whole. It is recommended to rework (the Abstract at least, if no Conclusions are present) to make it clear and self-sustained.*
>
> **Response:** This is a great point and we have revised accordingly. In particular, we have edited the abstract defining balancedness and bounded conditioning, so that it is now self-contained, see l. 21-24 of the revision. We have also added a final section titled ‘Discussion and Concluding Remarks’, where we start by presenting the main message of the paper, then provide a discussion and conclude with future directions.
>
>
>
>
>
> *W2: Quite a few assumptions are required in general. More over thay are added along the way, e.g., Assumption 4.2, $|\sigma(x)|\le |x|$, etc., (which is ok, but also narrows the applicability of the final result). Some of those are very technical (especialy those in Theorems, such as in Theorem 3.1, (2),(3), (4)) but as well Assumption 4.3, Theorem 4.4. (10) and more) and with paper specific notation to learn, e.g., $\lambda_{3\to L}$. It would help paper to have an thorough discussion on applicability/limitations of these assumptions. Perhaps at expense of shortening some 'proof sketches' refering them to SM?*
>
> **Response:** We agree with this and have revised accordingly. A detailed discussion on the assumptions is contained in the new Section 7 of the revision. In particular, we note that Theorem 3.1 provides a connection between neural collapse and properties of well-trained networks, i.e., approximate interpolation, approximate balancedness, bounded representations/weights and good conditioning of the features. As such, the result is rather general and requires no assumptions beyond the aforementioned properties. Theorem 4.4 then instantiates our framework for proving neural collapse to a class of networks with pyramidal topology (Assumption 4.1), smooth activations (Assumption 4.2) and for a class of initializations (Assumption 4.3). These assumptions are used only for the analysis of the first phase of the training dynamics, where the network achieves approximate interpolation. Thus, they could be replaced by any other set of assumptions guaranteeing that gradient descent reaches small training loss. Specifically, such guarantees are obtained by Zou & Gu (2019) for deep ReLU networks (with stronger requirements on over-parameterization but no assumptions on the topology) and by Bombari et al. (2022) for networks with minimum over-parameterization (under a requirement on the topology milder than Assumption 4.1). As concerns Assumption 4.3 on the initialization, we discuss in page 5 a setting where it holds. In addition, by following the argument in Appendix C of Nguyen & mondelli (2020), one readily obtains that Assumption 4.3 also holds for the widely used LeCun's initialization, i.e., $W_\ell^0$ has i.i.d. Gaussian entries with variance $1/n_{\ell-1}$ for all $\ell\in [L]$, as long as $n_1=\Omega(N)$.
>
> To make space and for the sake of readability, we have simplified the statements of Theorems 3.1 and 4.4 using a big-O notation, as suggested by Reviewer 1hwn.
>
>
> *W3: 'Discussion and Conclusion' sections are not presented. This is most likely due to space constrained and will be added in case of acceptance (?). Yet, I find it impacts the paper quality negatively. Especially in a light of the previous (W2) comments, the discussion and conclusions could have brought a critical 'back to reality' summary. Idealy it would bring more intuition for results and their application. Yet, I find it very important to have such Discussion reviewed before publishing …*
>
> **Response:** This is an excellent suggestion, and we have now added the new Section 7 titled “Discussion and Concluding Remarks”. There, we discuss our assumptions, provide a comparison with earlier work by Nguyen & Mondelli (2020), discuss connections with the NTK regime and conclude with future directions.

---

> > ### Author Response · Authors · 2024-11-18
> > **Response (2)**
> >
> > *W4: Some references are rather inaccurately interpreted/generalized too much perhaps. For instance the lines 399-400 "...Thankfully, instead of simply diverging, for large $\eta$ (but not too large) the parameters naturally end up at the ‘edge of stability’: the top eigenvalue of the Hessian is close to $2/\eta$, i.e., the threshold below which GD is stable..." from (Cohen et all. 2021). Referenced work provides only experimental evidence for a phenomenon and only approximately, i.e., for certain settings operator norm of Hessian exceeds 'stability threshold' , etc. Than approximation used on l410, especially  is only valid if $\nabla^2_\theta Z_L$ norm is bounded, which is ok for NTK regime, but not necessarily for large learning rate regime. Or is it?*
> >
> > **Response:** We are aware that the evidence for the edge of stability phenomenon is almost only empirical at this stage, and we will make it clearer in the main (and mention some already existing theoretical evidence for toy models, e.g. https://openreview.net/forum?id=p7EagBsMAEO). Basically our goal was to offer two possible assumptions under which the conditioning can be controlled, thus guaranteeing NC2 and NC3 in addition to NC1: global convergence, or stability under large learning rates. There is no theoretical approach to prove either of these assumptions today, but there is empirical evidence for the stability under large learning rates in many different architectural settings. We are reasonably hopeful that the learning rate stability could be proven in the future, though it would probably require some new proof techniques (in contrast we are less confident about global convergence, though we kept it as it is a more common assumption, even though it is very strong).
> >
> > The approximation of l410 can be proven rigorously under the assumption that the weights are bounded and the interpolation error $\epsilon_1$ goes to zero.
> >
> > We now go into more details at the end of Section 5.2 of the revision to explain our intuition better.
> >
> >
> >
> > *W5: Following up on W4, Proposition 5.3, and other Theorem combine NTK with large learning rate regime, which sounds dangerous. Also requirement on wide first layer, suggest a NTK limit case is required. Could authors clarify a bit more on this relation?*
> >
> > **Response:** That’s a very good observation. The reason this is not a problem is that we only need the learning rate to be ‘reasonably large’, i.e. of order $1/L_1$, which is small enough for the NTK regime to be stable (in the NTK regime, the learning rate has to be chosen as $1/||\Theta||\_{op}$ and the NTK $\Theta$ is a sum over the layers, so it typically scales linearly with depth). In contrast, for the moment our proofs require an extremely small learning which is negatively exponential in the depth $L_1$ (see Equation 27 in Theorem B.2), and it appears that these constraints actually come from the second part of training (after the NTK regime), where we only have a control on the parameter norm, and the worst case upper bound on the Hessian over bounded parameters is exponential in depth (Lemma C.1).
> >
> > We added a more detailed discussion of these aspects at the end of Section 5.2 of the revision.

---

> > > ### Author Response · Authors · 2024-11-18
> > > **Response (3)**
> > >
> > > *Q1: Line 188: "... approaches 1 ..." Shouldn't it be "... approaches 2" based on (3)? Is it still sufficient for orthogonality claims (can the proof be adjusted to account for it)?*
> > >
> > > **Response:** We believe our sentence in the text is correct and (3) really approaches 1. In fact, $\epsilon$ approaches 0 (as $\epsilon_2\to 0$) and $c_3^{1/L_2}$ approaches 1 (as $L_2\to\infty$), which implies that the overall expression evaluates close to 1.
> > >
> > > In the revision, we use a big-O notation in (3), which makes explicit the dependence on $\epsilon_2$. This should clarify that the expression approaches 1.
> > >
> > > *Q2: Proof sketch of Theorem 4.4., lines "306". Why and how are "two phases" guaranteed to happen during GD training?*
> > >
> > > **Response:** In the proof, the first phase relies on an NTK-type analysis to guarantee interpolation, and the second phase retains interpolation while the network becomes more and more balanced thanks to weight decay. This can also be viewed as a separation of timescales as $\lambda$ gets smaller: the number of steps needed to reach balancedness is of order $\frac{1}{\eta \lambda}$ (weight-decay timescale). This is significantly later than the interpolation time, which is of order $\frac{1}{\eta}$ (NTK timescale). From our proof, we know that the first phase is NTK-like, whereas the second phase is not, since we observe low-rank weight matrices and feature learning at the end of the second phase, which could not arise in a purely NTK regime.
> > >
> > > This strategy/dynamics of “NTK followed by weight decay” is to our knowledge novel and is an important theoretical contribution of our paper. It allows us to have the best of both worlds: interpolation from the NTK phase, and feature learning/balancedness from the second phase.
> > >
> > > We discuss this point in the paragraph ‘NTK regime and beyond’ added to the new Section 7 of the revision.

---

> ### Comment · Reviewer_H9Md · 2024-11-26
> **After rebuttal comments**
>
> Thank authors for a detailed revision and addressing raised concerns sufficiently. Especially discussion and clarifications (NTK + large learning rate regime as per original review) added to the paper are appreciated. Overall, amendments are significant and I raise my score to 6.

---

### Author Response · Authors · 2024-11-18
**General response**

We thank the reviewers for the overall positive evaluation of our work and for the detailed comments. We discuss them in detail in the separate responses to each reviewer, and we have uploaded a revision with main changes highlighted in red.

Following the suggestion of several reviewers, we have added a final section titled ‘Discussion and Concluding Remarks’, where we start by presenting the main message of the paper, then we discuss our assumptions, we compare with earlier work by Nguyen & Mondelli (2020), we discuss how our analysis relates to the NTK regime, and we conclude with a direction for future research.

To make space, as well as to improve the presentation of our results, following the suggestion of reviewer 1hwn, we have simplified the statements of Theorem 3.1 and Theorem 4.4: in the revision, we hide constants via a big-O notation and defer the precise statements to the appendix.

---

### Meta-Review · Area_Chair_8LpU · 2024-12-20

**Metareview:**

This paper explores neural collapse, a geometric structure observed in the last layer of deep neural networks (DNNs) at convergence, where training data is consistently represented. Moving beyond the unconstrained features model, the authors study DNNs with at least two linear layers and establish conditions for neural collapse, including low training error, balancedness of linear layers, and bounded conditioning of pre-linear features. They prove these conditions hold during gradient descent training with weight decay, particularly in networks with wide first layers and stable solutions. The authors claim that this work provides the first theoretical demonstration of neural collapse in end-to-end DNN training.

The reviewers raised the following strengths and weaknesses

Pros:

+ First to prove neural collapse in end-to-end DNN training without unconstrained features this paper introduces key conditions like balancedness and bounded conditioning.

+ This paper validates results with experiments on MNIST and CIFAR; aligns theory with training practices like gradient descent and weight decay.

+ The paper highlights the role of linear layers, large learning rates, and weight decay in achieving neural collapse.


Cons:

- Accessibility Issues: Dense initial presentation; abstract lacks clarity for general readers.

- Strong Assumptions: Relies on two linear layers, specific initialization, and pyramidal topology, limiting general applicability.

- Missing Aspects: No discussion of NC4; broader implications and applications were underexplored.

Some of these concerns were assuaged by the rebuttal of the authors and all reviewers are in favor of acceptance. I concur

**Additional Comments On Reviewer Discussion:**

Some of these concerns were assuaged by the rebuttal of the authors and all reviewers are in favor of acceptance.

---

### Decision · Program_Chairs · 2025-01-22

Accept (Oral)